# Can LLMs Learn by Teaching for Better Reasoning? A Preliminary Study

**Xuefei Ning**[*1], **Zifu Wang**[*2], **Shiyao Li**[*1,3], **Zinan Lin**[*4], **Peiran Yao**[*3,5],
**Tianyu Fu**[1,3], **Matthew B. Blaschko**[2], **Guohao Dai**[6,3], **Huazhong Yang**[1], **Yu Wang**[1]
[1]Tsinghua University [2]KU Leuven [3]Infinigence-AI
[4]Microsoft Research [5]University of Alberta [6]Shanghai Jiao Tong University

## Abstract

Teaching to improve student models (e.g., knowledge distillation) is an extensively studied methodology in LLMs. However, in human education, teaching enhances not only the students but also the teachers by fostering more rigorous and clearer reasoning, as well as deeper knowledge building. We ask: *Can LLMs also learn by teaching (LbT) for better reasoning?* If the answer is yes, we can potentially unlock the possibility of continuously advancing the models without solely relying on human-produced data or stronger models. In this paper, we provide a preliminary exploration of this question. We show that LbT ideas can be incorporated into existing LLM training/prompting pipelines and bring improvements. Specifically, we design three methods, each mimicking one of the three levels of LbT: observing students' feedback, learning from the feedback, and learning iteratively, with the goal of improving answer accuracy without training or improving models' inherent capability with fine-tuning. We reveal some findings: (1) *Teaching materials that make it easier for students to learn (via in-context learning) have clearer and more accurate logic*; (2) *Weak-to-strong generalization*: LbT might help improve strong models by teaching weak models; (3) *Diversity in students might help*: teaching multiple students could be better than teaching a single student or the teacher alone. We hope that our exploration can inspire future research on LbT and, more broadly, the adoption of advanced education techniques to improve LLMs. The code and website are at https://github.com/imagination-research/lbt and https://sites.google.com/view/llm-learning-by-teaching.

## 1 Introduction

> *I couldn't reduce it to the freshman level. That means we really don't understand it.*
>
> *– Richard Feynman*

"*Learning from teachers (LfT)*" is a common pipeline in machine learning, especially in the realm of Large Language Models (LLMs). For example, knowledge distillation [24, 44, 75] and distillation via synthetic data [1, 28, 41] focus on transferring the knowledge from teacher LLMs to student LLMs by letting teacher models *teach* student models through token logits, features, or synthetic data [85]. They become the go-to methods for closing the performance gap between open-source and proprietary LLMs, as well as for maintaining performance during model compression.

In fact, in human learning, *teaching not only benefits students but can also improve the teachers themselves*. "*Learning by teaching (LbT)*", also known as the Feynman learning method, is proven to improve human learning by fostering rigorous and clear reasoning as well as knowledge building [5, 18, 22, 33, 62–65]. Fig. 1 illustrates the conceptual comparison of the LfT and LbT pipelines.

---

[*]Equal contribution.
Corresponding to: foxdoraame@gmail.com (Xuefei Ning), zinanlin@microsoft.com (Zinan Lin), yu-wang@tsinghua.edu.cn (Yu Wang)

38th Conference on Neural Information Processing Systems (NeurIPS 2024).

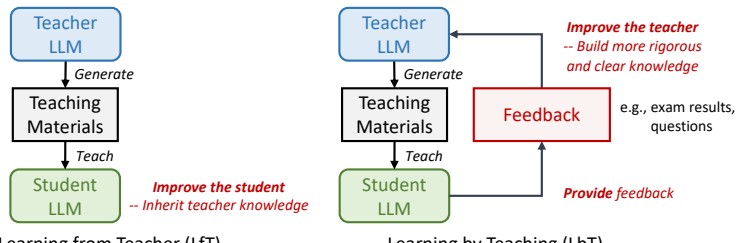

Figure 1: **Left:** *Learning from teacher* aims at improving student LLMs with knowledge from the teacher LLMs. It is the essential idea behind common approaches including knowledge distillation and distillation via synthetic data. **Right:** In contrast, *Learning by teaching* aims at improving *teacher LLMs* through the teaching process using feedback from student LLMs.

Table 1: The explored **M1**, **M2**, **M3** methods.

| LbT Level | Objective | Pipeline | LbT Implementation | Method Abbrev. |
|---|---|---|---|---|
| L1 | Improve the answer quality without training | Search-based output generation | Scoring based on students' performance | **M1** (§ 3) |
| L2 | Improve the inherent model ability with training | Generation-scoring-finetuning | | **M2** (§ 4) |
| L3 | Improve the answer quality without training | Input prompt optimization | Analyzing feedback from multiple students | **M3** (§ 5) |

Motivated by this insight, in order to improve one of the most crucial abilities of LLMs – the reasoning ability, we want to ask: *Can LLMs also learn by teaching for better reasoning?* In addition to improving reasoning, as one can imagine, LbT could open exciting opportunities for the models to *continuously evolve* by teaching other (potentially weaker) models, rather than solely relying on human-produced data or stronger teacher models. More broadly, we hope that this exploration could provide insights on borrowing advanced education techniques to improve LLMs [16, 32, 53].

To explore this question, we draw on learning science literature that connects LbT in human learning with reflection [5, 13, 48] and knowledge-building [62–64], summarizing three *levels* of LbT:

- **L1: Observing students' feedback.** The teacher instructs the students, who then provide feedback (e.g., taking exams and reporting the scores, asking questions about unclear logic).
- **L2: Learning from the feedback.** Based on the feedback, the teacher can analyze which logic and concepts the students might have (mis)understood. This information is useful for the teachers to improve their teaching strategy, and further enhance the teacher's own understanding of the concepts.
- **L3: Learning from the feedback iteratively.** The teacher can teach the students, observe the feedback (L1), and learn from the feedback (L2) *iteratively*.

In this paper, we study the viability of instantiating these LbT ideas in LLMs. There is a range of possibilities in terms of the objective, the pipeline, and the implementation (§ 2 and Tab. 1). As an initial exploration, we study three methods, each for one of the three LbT levels.

- **M1** aims at improving LLMs' answer quality by directly utilizing students' feedback (L1). More specifically, given a set of generated answers, we *score each rationale based on its ability to teach student models using in-context learning (ICL) to correctly answer similar problems*. We show that aggregating multiple rationales [79] with LbT-based scores can improve the answer accuracy. Notably, **M1** improves GPT-4o's accuracy on the MATH dataset [27] from 87.84% to 96.69%.
- **M2** aims at improving LLMs' inherent ability by learning from students' feedback (L2). We use the approach in **M1** to score teacher-generated rationales. Then, we apply direct preference optimization (DPO) [59] to fine-tune the teacher model with the rationale-score pairs. We show that **M2** is better than using DPO with correctness scores.
- **M3** aims at improving LLMs' answer quality by iteratively learning from students' feedback (L3). Specifically, we prompt the LLM to *reflect on the failure cases of multiple students* and *devise new positive and negative exemplars*. We show that the LLM can improve the exemplars based on feedback from multiple students. These improved exemplars used in prompts not only improve the learning outcomes for multiple students but also enhance the teacher's performance.

We reveal some interesting or promising findings related to LbT:

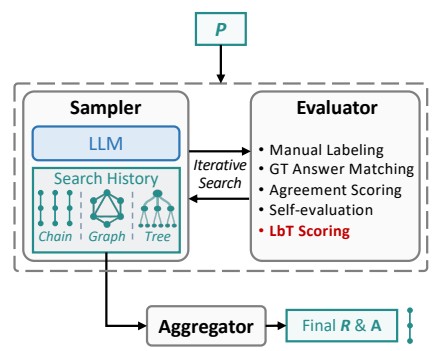

(a) The "search-based output generation pipeline" for improving the answer quality.

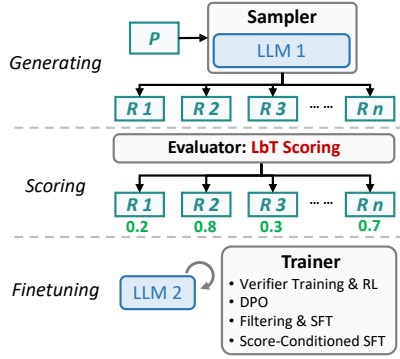

(b) The "generating-scoring-finetuning pipeline" for improving model capability.

Figure 2: Two general pipelines for improving the answer quality and model capability. "*P*" stands for "Problem"; "*R*" stands for "Rationale"; "*A*" stands for "Answer".

- **Teaching materials that make it easier for students to learn have clearer and more accurate logic (LbT-TMQ[1] assumption)** when using ICL as the student's "learning" method: Our LbT-based scoring relies on this assumption, and our results and inspection support this assumption.
- **Weak-to-strong generalization**: Strong teachers can improve even when teaching weaker students, suggesting some promise of using LbT to improve superhuman models [6].
- **Diversity in students might help**: Rather than teaching the teacher itself, teaching *other* students and *multiple* students might help. This suggests the feasibility of using LbT to synergize the capability and knowledge from multiple models.

To summarize, with appropriate pipelines and teacher-student settings, LbT can help improve LLMs' answer quality and inherent capability. We believe that these preliminary case studies are only scratching the surface of the potential of LbT. As LLMs are becoming increasingly powerful, more advanced approaches in pedagogy can potentially help with the inference and training of LLMs.

## 2 Related Work of Our Learning by Teaching Implementations

As shown in Tab. 1, we study two types of objectives: *improving answer quality without training* and *improving the inherent ability of the model with training*. § 2.1 and § 2.2 describe how **M1**, **M2**, and **M3** relate to prior work on these two objectives, respectively. See App. D.5 for more discussion.

### 2.1 Improving the Answer Quality without Training

Existing literature has incorporated various insights from the human reasoning process to develop prompting-based methods, including writing down the thinking process [36, 81], subproblem decomposition [52, 56, 94], fetching the abstract principles and answering based on them [93], self-reflection-based answer refinement [47, 67], and so on. We explore two ways of incorporating the LbT insight to implement two prompting-based methods:

- **M1** relates to the popular "search-based output generation pipeline" shown in Fig. 2a [4, 42, 45, 47, 67, 79, 84, 87]. This pipeline iteratively samples and evaluates new rationales or rationale steps for searching the optimal output, and ultimately derives the final rationale or answer from the search history. One essential component in this pipeline is an *evaluator* who evaluates the quality of each rationale or rationale step. We design an LbT evaluator that *scores each generated rationale based on its ability to teach student models to correctly answer similar problems*.
- **M3** relates to existing prompt optimization methods [57, 71, 95] that iteratively improve the prompts based on their performance (e.g., accuracy, failure cases). The key innovation in **M3** is how it evaluates the "performance": instead of evaluating with the same model that produced the prompts (i.e., the teacher model), we test how the prompt works with *other* student models and show that this change benefits the prompt tuning outcome.

### 2.2 Improving the Inherent Model Capability with Training

To improve the inherent model capability, **M2** incorporates the LbT insight into the "generating-scoring-finetuning pipeline". Fig. 2b illustrates the three steps in the pipeline: (1) Letting the target

---

[1]Refers to "teaching material quality". See App. D.2 for more discussion.

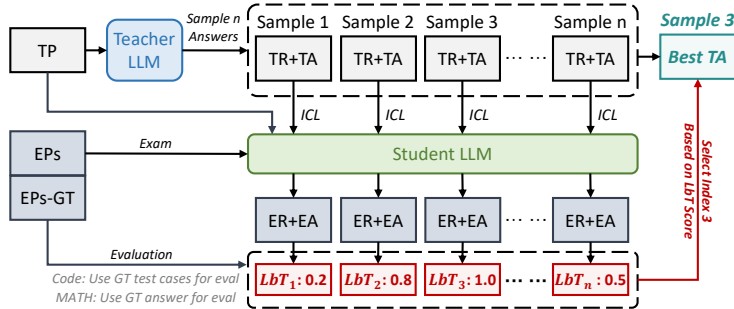

Figure 3: **M1**. The goal is to derive the best TA from the TR-TA pairs generated by the teacher LLM.

LLM or a teacher LLM generate multiple rationales for a given problem; (2) Scoring the rationales using an evaluator; (3) Utilizing the rationales and scores to (optionally) train a verifier [14, 45, 77], and finetune the target LLM by reinforcement learning [77], DPO [59, 88] or its variant [49, 55], filtering and supervised finetuning (SFT) [31, 89, 90], or score-conditioned SFT [43, 46].

In these works, the rationale scoring is usually achieved through manual labeling [45, 46, 99], ground-truth (GT) answer matching [89, 90], agreement-based scoring [31], or self-evaluation [88]. In contrast, **M2** *scores the rationale based on its ability to teach student models to correctly answer similar problems*. In this way, **M2** can provide *automatic* and *fine-grained* quality evaluation for rationales, which helps automate and improve the continual evolution of models' capability.

## 3 Method (M1) for LbT Level 1: Observing Students' Feedback

### 3.1 Method

One common teaching strategy in education is that the teacher first teaches students how to solve a class of problems by giving them the example rationale (named *Teaching Rationale*, or *TR* in short) and the answer (named *Teaching Answer*, or *TA* in short) to a particular question (named *Teaching Problem*, or *TP* in short). Then, the teacher asks students to solve other similar problems (named *Exam Problem*, or *EP* in short) to test if the students understand the concepts. The teacher can also learn from this process by observing the feedback (i.e., *LbT level 1*): if the students can answer EPs well, then it likely means that the TR-TA pair is of high-quality.

Our idea is to implement this strategy in LLMs to select high-quality TR-TA pairs. As depicted in Fig. 3 and Alg. A1, we first instruct the teacher model to solve a given TP multiple times, resulting in multiple TR-TA pairs. Then, each TR-TA pair is used as an in-context learning (ICL) example to guide the student model in solving a series of EPs. With the produced Exam Rationales (ERs) and Exam Answers (EAs), each student will then receive an exam score (i.e., the accuracy of EAs), denoted as the LbT score. The LbT score can be used as a quality assessment of the corresponding TR-TA pair. We consider two ways to select the final TA [79]: (1) We select the TR-TA pair with the highest LbT score. We denote this approach as "**M1 (MAX)**". (2) For TAs that can be aggregated via exact matching, such as mathematical reasoning, we can take the sum of the LbT scores for each TA separately and find the TA with the maximum sum. We denote this approach as "**M1 (SUM)**".

The following subsections present the evaluation of **M1** on mathematical reasoning and code synthesis tasks. Please refer to App. D for the rationale behind the task selection.

### 3.2 Evaluation on Mathematical Reasoning

#### 3.2.1 Experimental Setups

We use the extension MATH() [72] of the MATH dataset [27], where each problem has variants with different values. Following the train-test split specified by [45], among the 500 test problems, 181 problems are provided with 3 functional variants each. We use these 181 problems as TPs. For each TP, we sample 256 TR-TA pairs. Then, using each TR-TA pair as the ICL exemplar, we use the 3 functional variants of TP as EPs. Each exam is repeated 3 times with randomized student decoding, resulting in 9 ER-EA pairs. Each TA is scored based on the correctness of the 9 EAs.

#### 3.2.2 Results

We show the results in Tab. 2 and provide analyses as follows. More results are in App. A.

Table 2: Results on 181 MATH test problems with 256 TR-TA pairs. The best results of each row are highlighted in green. The "Improv" column calculates the improvements of average performance achieved by **M1** (SUM) over SC.

| Teacher | Student | Greedy | SC | **M1** (MAX) | **M1** (SUM) | Improv. |
|---------|---------|--------|-----|-------------|-------------|---------|
| GPT-4o | GPT-4o mini | 87.84 | 91.71 | 95.03 | 96.69 | +4.98 |
| GPT-4o | LLaMA3-8B | 87.84 | 91.71 | 94.48 | 95.03 | +3.32 |
| GPT-4o | GPT-4o mini & LLaMA3-8B | 87.84 | 91.71 | 96.13 | 95.58 | +3.87 |
| GPT-3.5 | LLaMA3-8B | 59.11 | 77.90 | 83.43 | 83.43 | +5.53 |
| GPT-3.5 | Mistral-7B | 59.11 | 77.90 | 81.22 | 83.43 | +5.53 |
| GPT-3.5 | LLaMA3-8B & Mistral-7B | 59.11 | 77.90 | 84.53 | 84.53 | +6.63 |
| LLaMA3-70B | LLaMA3-8B | 70.16 | 81.77 | 86.74 | 87.85 | +6.08 |
| LLaMA3-70B | Mistral-7B | 70.16 | 81.77 | 86.19 | 85.08 | +3.31 |
| LLaMA3-70B | LLaMA3-8B & Mistral-7B | 70.16 | 81.77 | 87.85 | 87.29 | +5.52 |
| LLaMA3-8B | LLaMA3-8B | 45.85 | 64.64 | 77.90 | 82.87 | +18.23 |
| Mistral-7B | LLaMA3-8B | 19.88 | 40.88 | 51.93 | 53.59 | +12.71 |

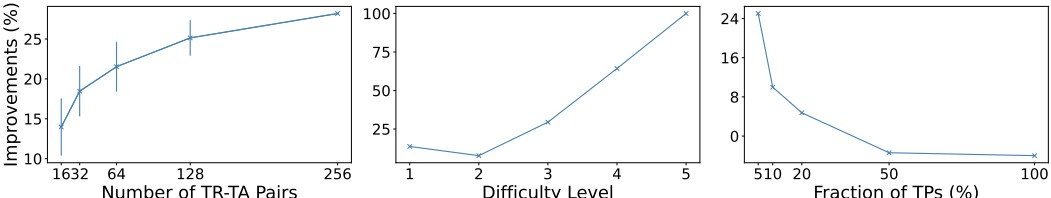

Figure 4: Relative improvements of **M1** over SC using LLaMA3-8B as the teacher and student on 181 MATH test problems with respect to: (**Left**) Number of TR-TA pairs. Error bars are calculated using the bootstrap sampling technique [43], where 10 subsets are sampled from the 256 TR-TA pairs, and standard deviations are computed across these sets; (**Middle**) Difficulty level; (**Right**) The fraction of TPs when sorted by the cosine distance to the 2 closest problems from the training set.

**M1 is effective with various model settings and surpasses baselines**. **M1** exceeds self-consistency (SC) [79] with various model settings: strong-teach-weak (e.g., GPT-4o teaches GPT-4o mini), weak-teach-strong (e.g., Mistral-7B teaches LLaMA3-8B), and self-teaching (e.g., LLaMA3-8B teaches itself). **M1** (SUM) outperforms **M1** (MAX) in most cases. We also show that LbT-based scoring surpasses self-evaluation scoring [35, 74, 84, 88] in Tab. A7. Since **M1** incurs higher inference cost than SC when using the same number of TR-TA pairs, we also conduct an experiment in Tab. A6, showing that with comparable or much lower compute, **M1** with just 24 TR-TA pairs achieves a 0.17%∼8.29% accuracy improvement over SC with 256 TR-TA pairs.

**M1 can further benefit from multiple students**. Using GPT-3.5 to teach both LLaMA3-8B and Mistral-7B achieves a significant improvement than teaching LLaMA3-8B or Mistral-7B separately.

**M1 can identify infrequent but correct TAs**. **M1** can efficiently discover the correct answer from many teacher samples, whereas SC requires the correct answer to be in the majority to derive it. Fig. 4 (left, middle) shows the improvements of **M1** over SC across different numbers of TR-TA pairs and difficulty levels. The relative improvement of **M1** over SC increases as the number of TR-TA pairs or the difficulty levels grow within the experimental range.

**The TP and the corresponding EPs should be similar**. It is crucial to choose EPs similar to a TP such that the student can apply the logic from TR to solve EPs. We use the functional variants as EPs, which are very similar to TPs. To verify the necessity of TP-EPs similarity, we conduct an experiment that selects similar EPs from the original MATH training set. We calculate the embedding of each TP using the "all-mpnet-base-v2" sentence embedding model [60], and select the 2 closest problems from the training set as EPs. We sort TPs by the cosine distance to the corresponding EPs and calculate the relative improvements over SC on a fraction of TPs. Fig. 4 (right) shows that **M1** only provides improvements for TPs that have similar problems in the training set.

### 3.3 Evaluation on Competition-Level Code Synthesis

#### 3.3.1 Experimental Setups

We use the Grandmaster Dynamic Programming (DP) study plan on LeetCode.[2] Each dataset in the study plan has 5∼10 problems, and each problem has 2∼3 visible test cases and many hidden

---

[2]https://leetcode.com/studyplan/dynamic-programming-grandmaster/

Table 3: **S-score** results on *Game Theory* dataset in LeetCode Grandmaster DP study plan. "SG-1"-"SG-4" and "PW" are abbreviations of individual questions in the dataset; see Tab. A8 for details. The results of **M1** that improve (degrade) by more than 0.01 are highlighted in green (red).

| Models | Metrics | SG-1 | SG-2 | SG-3 | SG-4 | PW |
|---|---|---|---|---|---|---|
| T=LLaMA3-8B S=LLaMA3-8B | Avg. | 0.215 | 0.004 | 0.216 | 0.604 | 0.609 |
| | **M1** (MAX) | 0.630 | 0.004 | 0.228 | 1 | 0.508 |
| | Avg. (V-score=1) | 1 | - | - | 0.755 | 0.851 |
| | **M1** (MAX) (V-score=1) | 1 | - | - | 1 | 1 |
| T=LLaMA3-8B S=LLaMA3-8B (w. Self-Debugging) | Avg. | 0.348 | 0.004 | 0.319 | 0.608 | 0.694 |
| | **M1** (MAX) | 0.348 | 0.011 | 0.570 | 0.771 | 0.746 |
| | Avg. (V-score=1) | 0.797 | - | - | 0.722 | 0.851 |
| | **M1** (MAX) (V-score=1) | 1 | - | - | 1 | 0.935 |
| T=GPT-3.5 S=GPT-3.5 | Avg. | 0.582 | 0.007 | 0.428 | 1 | 0.645 |
| | **M1** (MAX) | 1 | 0.011 | 0.681 | 1 | 1 |
| | Avg. (V-score=1) | 0.994 | - | 0.714 | 1 | 0.894 |
| | **M1** (MAX) (V-score=1) | 1 | - | 0.135 | 1 | 1 |
| T=GPT-3.5 S=GPT-3.5 (w. Self-Debugging) | Avg. | 0.701 | 0.133 | 0.592 | 1 | 0.853 |
| | **M1** (MAX) | 1 | 0.337 | 0.714 | 1 | 0.968 |
| | Avg. (V-score=1) | 0.996 | 1 | 0.714 | 1 | 0.911 |
| | **M1** (MAX) (V-score=1) | 1 | 1 | 0.714 | 1 | 0.968 |
| T=LLaMA3-70B S=LLaMA3-8B | Avg. | 0.875 | 0.008 | 0.679 | 1 | 0.601 |
| | **M1** (MAX) | 1 | 0.007 | 1 | 1 | 1 |
| | Avg. (V-score=1) | 1 | - | 1 | 1 | 0.883 |
| | **M1** (MAX) (V-score=1) | 1 | - | 1 | 1 | 1 |

test cases. We assign a visible score (**V-score**) of 1 and 0 to the code that *passes all* or *fails any* visible cases [43]. To evaluate the actual correctness of a code, we submit the code to LeetCode, and record the pass rate on the hidden cases as the submit score (**S-score**). For a TP, we sample 8 TR-TA pairs from the teacher, where TR is a rationale in natural language, and TA is a Python code (See Ex. 1 for an example). Each TR-TA pair is assigned an LbT score by teaching a student to solve the remaining problems in the dataset. **M1** calculates the exam **V-score** as the LbT score to avoid additional LeetCode submissions. Check App. A.3.2 for additional setups.

### 3.3.2 Results
Here, we analyze the results on the Game Theory dataset. Check App. A.3 for additional results.

**M1 can be more general than agreement-based methods such as SC. M1** (MAX) does *not* require an oracle to assess the equivalence of two answers, which is challenging for codes. Therefore, we only use the average pass rate (with or without V-score=1 filtering) as the baseline. Nevertheless, when such an oracle is provided [8, 43, 66], we can use **M1** (SUM) which was shown to be better than **M1** (MAX) in § 3.2 . We defer this exploration to future work.

**M1 selects better TR-TA than the baseline in most cases.** If the student closely follows the strategies in TR-TA to solve EPs, the student exam score can indicate the quality of TR-TA. (1) When the TR-TA has high quality (Ex. 1), the student mimics the teacher's strategy to solve the EP with a correct DP code. (2) When the TR-TA is logically incorrect, e.g., DP code with wrong recurrences (Ex. 2) or a non-DP wrong code (Ex. 3), the student also follows the wrong TR-TA with a wrong ER and EA. (3) When the TR-TA is logically correct but has high complexity, e.g. recursive re-computation instead of DP (Ex. 4), the student also writes a recursion with high complexity.

As shown in Tab. 3, using the **V-score** on the few visible test cases can filter out some low-quality code, but **M1** can identify better TA in most cases. This is because LbT-based scoring can leverage student scores on similar EPs, providing a more informative evaluation of TA. Note that **M1** shows the largest improvements on TPs with medium difficulty. For very simple (e.g., SG-4 for GPT-3.5) or challenging (e.g., SG-2) problems, **M1** shows marginal or no improvements.

**Self-Debugging (SD) is both complementary to and beneficial for M1**. We experiment with applying one-iteration SD [11] using Prompt 5. Applying SD on TAs can provide **S-score** benefits complementary to **M1**, since SD fixes simple non-logical bugs, such as missing imports, miswritten variable names, and incorrect usage of library functions (an example is shown in Ex. 6), whereas **M1** mainly assess the quality of the logic. In addition, applying SD on EAs leads to more informative LbT score, as fixing non-logical bugs can make the students' exam **V-score** more indicative of quality of the TR-TA. Tab. 3 shows that after incorporating SD for both **M1** and the baselines, **M1** achieves consistent improvements.

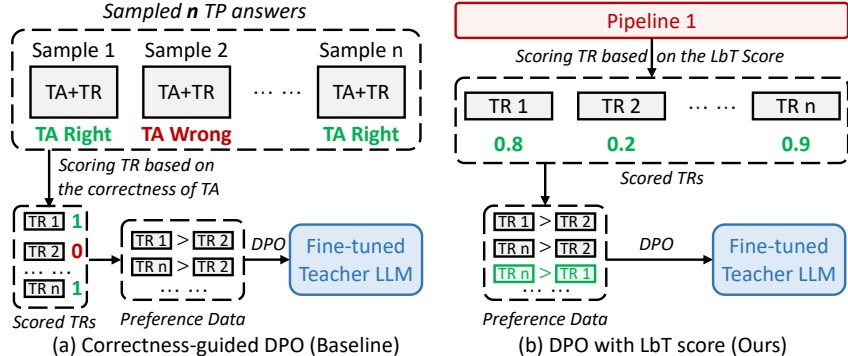

Figure 5: Baseline vs. **M2**. Both approaches use *scores* of TRs to craft preference data and finetune the teacher LLM with DPO. **Left:** The correctness score of TA. **Right:** The LbT score of TR and TA.

Table 4: Results on 500 MATH test problems with greedy decoding.

| Teacher/Student | Original | Correctness-DPO | **M2** |
|---|---|---|---|
| LLaMA3-8B | 29.0 | 30.4 | 32.2 |

**For competition-level code synthesis task, M1 is more effective when the teacher and student come from the same family**, as shown by Tabs. 3 and A10. We find that this is because the student can follow a teacher from the same model family better, making the feedback more informative. A failure case of student-following when GPT-3.5 teaches LLaMA3-8B is shown in Ex. 5.

**TPs and EPs should be similar**. Most failure cases in Tabs. 3 and A10 occur when solving the "PW" TP. We find that this is because the solving of "PW" involves 2D DP, which differs from other problems that can be solved with 1D DP. Consequently, the student cannot follow TA to solve EPs.

## 4    Method (M2) for LbT Level 2: Learning from the Feedback

### 4.1    Method

In education, after identifying which teaching materials (e.g., TR-TA pairs) can enhance student performance (§ 3.1), teachers can use this information to improve their knowledge or teaching strategies. For example, if students perform poorly due to unclear or inaccurate teaching materials, teachers can correct their knowledge and avoid generating similar TR-TA pairs in the future.

We use this idea to train the LLMs to improve its reasoning ability. As depicted in Fig. 5, since the LbT-based scoring provides informative feedback on the quality of a TR-TA pair (verified in § 3), we collect the LbT scores of many TR-TA pairs and use them to finetune the teacher with DPO [59].

### 4.2    Experimental Setups

We use 1564 training problems from MATH() [72] as TPs. For each TP, we sample 32 TR-TA pairs from the teacher. For each TR-TA pair, we calculate $0.5 \times$ correctness score $+ 0.5 \times$ LbT score as its final score, where the correctness score is 1 or 0 when the corresponding TA is correct or wrong, respectively. For running DPO, we select pairs from the 32 TR-TA pairs whose score difference exceeds a threshold of 0.3, and keep at most 8 pairs of TR-TA pairs for each TP.

### 4.3    Results

Tab. 4 shows that **M2** achieves better results compared to solely using the correctness scores in DPO. This improvement is because LbT provides more informative scores than those purely based on correctness. One example is shown in Ex. 10. Although both TRs produce a correct TA, the losing TR is unnecessarily verbose and cannot be generalized to other similar problems. Another example is in Ex. 11. Although both TRs produce a wrong TA, the winning TR is logically better than the loser. LbT can discern the correct preference between these TR-TA pairs, thereby improving DPO results.

## 5    Method (M3) for LbT Level 3: Learning from the Feedback Iteratively

### 5.1    Method

We have shown that the students' exam scores can serve as an indicator of the ***reasoning quality*** of the teaching rationales. This indicator can be leveraged to aggregate better answers in **M1** and to further fine-tune the teacher in **M2**. In **M3**, we explore whether reflecting on students' detailed exam responses can help the teacher *iteratively* refine its teaching materials. Notably, we aim to

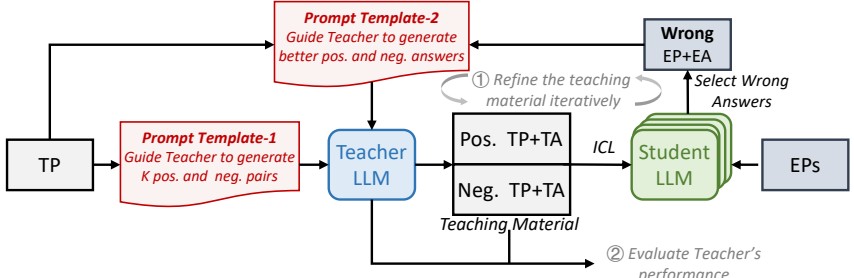

Figure 6: Overview of **M3**. The teacher teaches the students through a set of positive and negative ICL examples. These examples are iteratively refined by the teacher according to students' feedback.

Table 5: Teacher's $F_1$ score of **M3** on combined Liar dev and test set at the end of iteration $T$, where LLaMa3-70B is used as the teacher for all settings. The best results are in **bold**.

| Student(s) | $T = 1$ | $T = 2$ | $T = 3$ | $T = 4$ | $T = 5$ |
|---|---|---|---|---|---|
| LLaMa3-70B | 61.08±1.29 | 62.01±1.12 | 64.48±1.20 | 65.40±0.67 | 63.96±1.19 |
| LLaMa3-8B | 62.24±1.30 | **66.15±0.56** | **65.66±0.72** | 64.78±0.89 | 65.41±0.75 |
| LLaMa3-{70,8}B + Mistral-7B | **63.66±1.48** | 64.47±0.90 | 65.47±1.01 | **66.24±0.56** | **67.09±0.56** |

verify whether these refinements can enhance the teacher's own performance by providing more effective ***knowledge***. If so, we can assert that the iterative process of teaching, reflection, and material refinement facilitates some form of "knowledge building" [62–64] for the teacher. Additionally, we are interested in whether having *multiple* and *diverse* LLMs as students offers further benefits.

Specifically, we guide the teacher to iteratively improve teaching materials in the form of a set of positive and negative exemplars, based on the *student and teacher performance* when the set is used as the ICL examples. As depicted in Fig. 6 and Alg. A2, given a classification task, we first sample $K = 8$ positive and negative exemplars from the teacher, and then run multiple refinement iterations. Finally, we report the *teacher performance on the test set* when using the resulting ICL examples.

Each iteration contains the following steps: (1) *The current exemplars are used as the ICL examples to teach students to answer a set of EPs*. The EPs are randomly sampled from the training data in each iteration. (2) We select the EPs that students answered incorrectly and prompt the teacher to reflect on why the current exemplars might have misled students in these instances. (3) Based on the reflection, the teacher generates multiple updated exemplar sets. (4) We keep the exemplar set that *achieves the best teacher performance on the training data* when the set is used as the ICL examples.

## 5.2 Experimental Setups

We evaluate **M3** on two binary text classification tasks: Liar [78] and Logical Fallacy [34]. Liar is a dataset for false statement detection in political media, with 4,574 statements with speaker and context information. Logical Fallacy is a dataset of 2,449 samples of 13 logical fallacy types, which we adapt to classify the most common type *faulty generalization* against the rest of the types. We report the teacher $F_1$ score on the dev and test splits combined. Within an iteration, we choose the exemplar set with the highest teacher $F_1$ score on the training set. Across 14 random experiments, we report the mean $F_1$ and the standard deviation. We run a total of five refinement iterations.

## 5.3 Results

As Tabs. 5 and A17 shows, it is feasible to apply LbT on iterative prompt optimization: LLMs are able to reflect on the failure cases of students and propose revised exemplars that improve the teacher's performance, similar to the case of iteratively optimizing task descriptions as in previous work [57].

More importantly, we observe a performance gain brought by having *dedicated* students (as opposed to using a single LLM in prompt optimization as in previous work). Comparing to the scenario where the teacher and student are the same, having one or multiple LLMs different to the teacher as the student improves the quality of the teaching material faster. This demonstrates LbT as a case of *weak-to-strong* generalization. We speculate that the benefits are brought by more diverse error types made by a different (weaker) student model; see App. C.3 for more examples and analyses.

## 6 Broader Discussion

### 6.1 Insights into In-Context Learning

Currently, we conduct student "learning" with ICL, based on the assumption that students can effectively "learn" from ICL examples and apply similar strategies to solve EPs. Interestingly, prior

work [50] found that a correct input-output pairing in ICL examples does not matter much. At first glance, this finding seems to challenge our design, as it suggests that the TA accuracy may not affect the EA accuracy, which means the LbT score cannot reflect the quality of TR+TA. However, we find that, as opposed to only providing *labels* in the ICL examples [50], providing *rationales* is important. *LLMs can follow the problem-solving logic in the **detailed rationale** in the ICL examples well.* This may be because the rationale provides more information, making the ICL examples easier to follow. Consequently, we see that students can use similar logic as the TR when solving the EP. This means that better TR+TA can indeed lead to improved ER and thus higher EA accuracy (i.e., LbT score).

Our findings highlight two key factors for successful ICL following and for establishing a positive correlation between EA accuracy and TR quality/TA accuracy: (1) similarity between TP and EP, and (2) the use of Chain-of-Thought (i.e., detailed rationale). See App. A.3.5 for the effect of natural language rationale on ICL following in code synthesis. Note that we tried explicitly instructing the student to follow the ICL example, but it did not work well. We hope these findings can complement the current understanding of ICL [26, 40, 50, 83] and offer insights for improving ICL.

### 6.2 Weak-to-Strong Generalization

Improving models with human-generated/annotated data or synthetic data from stronger models is the dominant paradigm. However, how can we continuously improve the strongest model without relying on human-generated and annotated data? A recent work [6] conducts an exploration on using weak model supervision to train a larger model. Our work is another attempt towards the "weak-to-strong generalization" prospect by drawing from how humans continuously acquire new knowledge without direct instruction. We demonstrate that stronger models can further improve their own results (**M1**), parameters (**M2**), and prompt (**M3**) by utilizing the feedback of weaker models.

### 6.3 Limitations and Near-Term Extensions

**M1 and M2 rely on generating/selecting similar EPs.** We verify that LbT-based scoring can help select high-quality TR-TAs but require *the TP and EPs having similar problem-solving strategies*. In our experiments, suitable EPs are selected according to human-provided information in the dataset. One extension is to let a model automatically identify EPs similar to a TP from a large pool (Fig. 4). Another direction is to synthesize similar problems based on a group of problems and exploit the LbT principle to score many rationales for the new problems. Specifically, as a "self-instruct" [80] extension to **M2**, we can generate a new problem $P$ based on a group of problems $S = \{P_1, \cdots, P_k\}$ that are already known to be similar. The generating-scoring pipeline can then be applied to $P$ to obtain rationale-score pairs, where the LbT score can be easily obtained using $S$ as the EPs.

**Additional inference cost.** *LbT-based scoring in M1 and M2 requires additional inference cost*, which aligns with recent studies that show that increasing inference cost might be a promising way to improve models' reasoning capabilities [54, 69]. Nevertheless, designing efficient inference algorithms and systems [96] is needed to make these approaches more usable.

See App. E for other extensions and App. F for the discussion on potential risks of bias perpetuation.

### 6.4 Borrowing Education Strategies to Improve LLMs

**Borrowing the design strategies of teaching materials.** We show an LbT pipeline in Fig. 7. Each iteration involves six steps: (1) The teacher generates the Teaching Material (TM). (2) The student learns from the TM. Our work uses in-context learning for all student learning, but exploring other learning strategies is an interesting future direction. (3) The student provides feedback. The feedback can take many forms as listed in the figure. Our work mainly explored feedback in the form of exam details and scores. (4) The teacher reflects on the feedback and identifies the knowledge gaps in the TM or in the teacher's own knowledge. (5) The teacher can optionally refer to some external data source to address its own knowledge gaps. (6) The teacher improves the rigorousness, clarity, and completeness of their knowledge and updates the TM for the next iteration.

On one hand, updating TM can improve the teacher's own knowledge. For example, **M3** saves the updated exemplars as the teacher's prompt to improve the teacher's reasoning. On the other hand, a high-quality TM helps students learn better, so that the students can provide more meaningful feedback for the teacher. To create high-quality TM, it might be beneficial to borrow from TM design strategies in human education. Fig. 7 summarizes various TM design strategies, among which our work has explicitly explored three types (marked in black).

**Borrowing the pipelines.** We can borrow insights from education pipelines to design LLM inference and training pipelines. For example: (1) **Task-oriented collaborative multi-agent learning** [51]:

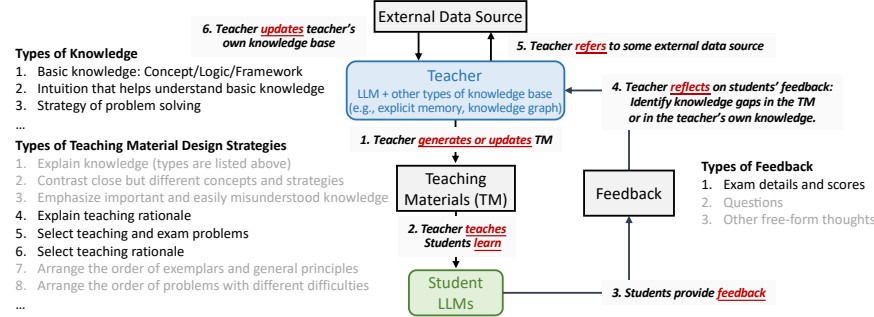

Figure 7: LbT pipeline and the summary of knowledge types, TM design strategies, and feedback.

Multiple LLM agents can form a collaborative study group to learn difficult topics in a task-oriented manner (see some discussions in App. D.2). Similar multi-agent collaboration ideas have been leveraged by LLM agent research [12, 17, 29, 39, 58, 91]. (2) **Better LbT by configuring proper teacher&student**: Literature on "teachable agents" finds that configuring a student's knowledge level appropriately can lead to more useful feedback for the teacher [5, 65]. This suggests the intriguing possibility of prompting the student LLM to "confine" its knowledge level [33], thereby amplifying the benefits of LbT for the teacher model. Furthermore, a junior model $\mathcal{M}$ can first teach a "student" that is stronger than itself, who can understand and critique mistakes and ambiguity. As $\mathcal{M}$ becomes stronger, it might become better at evolving its knowledge by teaching weaker students. This can be seen as a form of easy-to-hard task progression in curriculum learning [3]. (3) **Flexible teaching quality evaluation**: In human learning, feedback can take many forms [25, 82] beyond traditional exams [62–64], such as peer recommendations [48] and satisfaction questionnaires [37, 61]. Such mechanisms can be adapted to LLMs, potentially useful for open-ended tasks.

## 7 Conclusion

Aiming to improve LLM reasoning, we conduct a preliminary exploration of whether LLMs can "learn by teaching"— a well-known paradigm in human learning. We implement the LbT idea into well-established pipelines to develop three methods, and evaluate whether they improve reasoning performance on complex tasks such as mathematical reasoning and competition-level code synthesis:

- **M1** is based on the LbT-TMQ assumption. Specifically, we adopt ICL as the instructional method and measure the students' success in grasping the logic embedded in ICL examples by evaluating their performance on similar EPs. **M1** is implemented as a standard "search-based output generation" pipeline with an LbT-based rationale scoring component.

  **Results**: In mathematical reasoning, **M1** achieves a 3.31%∼18.23% accuracy improvement over the competitive SC baseline on 181 MATH test problems with 256 TR-TA pairs (Tab. 2). Note that **M1** achieves a 3.32%∼4.98% improvement on the powerful GPT-4o, reaching a high accuracy of 96.69%. Using comparable or much lower compute, **M1** with 24 TR-TA pairs achieves a 0.17%∼8.29% accuracy improvement over SC with 256 TR-TA pairs (Tab. A6).

  In code synthesis, **M1** achieves notable improvements in submission score in most scenarios, particularly when the teacher and student belong to the same model family (Tabs. 3, A11 and A13).

- **M2** uses the LbT scores from **M1** to fine-tune the teacher with DPO. **M2** is implmented as a standard "generating-scoring-finetuning" pipeline with an LbT-based rationale scoring component.

  **Results**: In the experiment of fine-tuning LLaMA3-8B, the **M2**-tuned model achieves a 1.8% accuracy improvement over the model tuned with correctness-based DPO, evaluated on 500 MATH test problems with standard greedy decoding (Tab. 4).

- **M3** lets the LLM iteratively refine ICL examples by analyzing the students' feedback.

  **Results**: For two binary text classification tasks requiring common-sense and logic reasoning, **M3** can craft better ICL examples through multiple refinement rounds, and the feedback from students other than the teacher itself is beneficial (Tabs. 5 and A17).

In summary, the LbT idea is implemented as a scoring method in **M1** and **M2**, and as an iterative refining pipeline in **M3** (Tab. 1). Our results suggest LbT's potential for harnessing the diversity offered by different students and facilitating weak-to-strong generalization in improving reasoning.

We believe our work only scratches the surface of leveraging educational principles to improve LLMs. Will these approaches find greater use as LLMs grow more intelligent? We discuss our research rationale and roadmap for exploring this intriguing question in App. D and § 6.

## Acknowledgement

This work was supported by National Natural Science Foundation of China (No. 62325405, 62104128, U19B2019, U21B2031, 61832007, 62204164), Tsinghua EE Xilinx AI Research Fund, and Beijing National Research Center for Information Science and Technology (BNRist). We thank for all the supports from Infinigence-AI. Z.W. and M.B.B. acknowledge support from the Research Foundation - Flanders (FWO) through project numbers G0A1319N and S001421N, and funding from the Flemish Government under the Onderzoeksprogramma Artificiële Intelligentie (AI) Vlaanderen programme. Z.W. and M.B.B. acknowledge LUMI-BE for awarding this project access to the LUMI supercomputer, owned by the EuroHPC JU, hosted by CSC (Finland) and the LUMI consortium, and EuroHPC JU for awarding this project access to the Leonardo supercomputer, hosted by CINECA. We thank Sergey Yekhanin from Microsoft Research for their support and suggestions for the work. We thank Zixuan Zhou, Chao Yu, Boxun Li for their discussions. We thank the anonymous reviewers for their insightful suggestions.

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

# Appendix

## Table of Contents

# A M1

## A.1 Workflow

The pseudo-code for M1 is illustrated in Alg. A1. In each iteration of the for loop, we calculate the LbT score of one TA in the following three steps:

1. Use teacher LLM to sample TR-TA pairs: As shown in line 4 of the Alg. A1, given a TP, the teacher LLM samples diverse TR-TA pairs as the teaching materials. The prompts of the teacher are shown in Prompt 1 (MATH) and Prompt 3 (Coding).

2. Use student LLM to take the exam: As shown in lines 7-11 in Alg. A1, the sampled TP-TR-TA pair serves as an in-context demonstration and we prompt the student LLM to solve related EPs. The prompts of the student LLM are shown in Prompt 2 (MATH) and Prompt 4 (Coding). Finally, we evaluate each EA and store its exam score to calculate the LbT score.

3. Calculate the LbT score for TA: As illustrated in lines 14-18 in Alg. A1, we use the exam scores of EPs to calculate the LbT score for each TA. For MATH, we employ both the MAX and SUM modes, while for Coding, we use only the MAX mode.

Finally, after getting the LbT score for all TAs, we select the TA with the largest LbT score.

---

**Algorithm A1** The Workflow of **M1**

---

**Require:**
    Teacher and Student LLM: **T**, **S**
    Teaching Problem: **TP**
    Exam Problems: **EPs**
    Ground-truth of Exam Problems: **EPs_GT**
    Number of TRs: **n**
    LbT Mode: **mode**
 1: lbt = defaultdict(float)
 2: **for** i = 1 to n **do**
 3:    # Teacher LLM samples TR and TA
 4:    TR, TA = **T**(TP)
 5:
 6:    # Student LLM performs exams
 7:    exam_scores = []
 8:    **for** EP, EP_GT in zip(EPs, EPs_GT) **do**
 9:      ER, EA = **S**(TP, TR, TA, EP)
10:      exam_scores.append(**Eval**(EA, EP_GT))
11:    **end for**
12:
13:    # Calculate the MAX/SUM LbT score of each TA
14:    **if** mode == "MAX" **then**
15:      lbt[TA] = max(lbt[TA], average(exam_scores))
16:    **else**
17:      lbt[TA] += average(exam_scores)
18:    **end if**
19: **end for**
20:
21: **return** max(lbt, key=lbt.get)

---

## A.2 Mathematical Reasoning

### A.2.1 Prompt Design

For MATH, we use 4-shot examples from Minerva [38]. The prompt for the student is the same as the teacher, except that an additional shot from the teacher (i.e. a TP-TR-TA pair) is appended after the original 4-shot examples.

> **Prompt 1. Teacher Prompt (MATH)**
>
> **[User:]**
> Your task is to answer the last question below. Give step by step reasoning before you answer. When you're ready to answer, please wrap your answer and conclude using the format
> """
> [[Final Answer]]:
> $ANSWER$
> """
>
> {4-shot examples}
>
> [[Question]]:
>
> {TP}
>
> **[Assistant:]**
> [[Solution]]:
> Let's think step by step.

> **Prompt 2. Student Prompt (MATH)**
>
> **[User:]**
> Your task is to answer the last question below. Give step by step reasoning before you answer. When you're ready to answer, please wrap your answer and conclude using the format
> """
> [[Final Answer]]:
> $ANSWER$
> """
>
> {4-shot examples}
>
> [[Question]]:
>
> {TP}
>
> [[Solution]]:
>
> {TR}
>
> [[Final Answer]]:
>
> {TA}
>
> [[Question]]:
>
> {EP}
>
> **[Assistant:]**
> [[Solution]]:
> Let's think step by step.

### A.2.2 Additional Experimental Setups

In § 3.2.1, both the teacher and the student use the same sampling parameters. Following [73], we use top-K sampling with K=20 and a temperature of 0.7.

In Tab. A7, we provide additional results in small-scale experiments. Specifically, for each of the 7 categories in MATH, we select 10 problems, resulting in a total of 70 TPs. We then sample 16 TR-TA pairs for each TP. The rest of the procedure follows § 3.2.1.

### A.2.3 Additional Results

**M1** requires additional inference costs for the student, raising concerns about whether **M1** can surpass the baseline within the same budget. We provide additional results in Tab. A6. Notably, using only 24 TR-TA pairs in **M1** still outperforms SC with 256 TR-TA pairs, especially on the most recent LLaMA3 models. Under this setting, **M1** has a lower inference cost than SC, particularly when comparing **M1** and SC on LLaMA3-70B and using LLaMA3-8B as the student (the first row of the table).

Furthermore, as shown in § 3.2.2, the improvement over the baseline does not saturate and actually increases, indicating that **M1** could achieve a higher upper-bound performance.

In Tab. A7, we compare LbT-based scoring with self-evaluation scoring [35, 74, 84, 88]. Our method consistently outperforms the self-evaluation baseline. Additionally, the table reveals that the performance gap between the strong-teach-strong and strong-teach-weak settings is relatively small.

Table A6: Results on 181 MATH test problems. SC is with 256 TR-TA pairs, while **M1** is with 24 TR-TA pairs. Standard deviations are calculated using the bootstrap sampling technique [43], where size-24 subsets are sampled from the 256 TR-TA pairs as the TR-TA set for **M1**, and standard deviations are computed across 10 sets. The "Improv" column calculates the improvements of average performance achieved by **M1** (SUM) over SC.

| Teacher | Student | Greedy | SC | **M1** (MAX) | **M1** (SUM) | Improv. |
|---|---|---|---|---|---|---|
| GPT-4o | GPT-4o mini | 87.84 | 91.71 | $94.20 \pm 0.79$ | $94.36 \pm 0.88$ | +2.65 |
| GPT-4o | LLaMA3-8B | 87.84 | 91.71 | $93.92 \pm 0.92$ | $94.14 \pm 0.83$ | +2.43 |
| GPT-4o | GPT-4o mini & LLaMA3-8B | 87.84 | 91.71 | $93.98 \pm 0.58$ | $94.31 \pm 0.43$ | +2.60 |
| GPT-3.5 | LLaMA3-8B | 59.11 | 77.90 | $78.34 \pm 1.86$ | $79.50 \pm 2.13$ | +1.60 |
| GPT-3.5 | Mistral-7B | 59.11 | 77.90 | $77.85 \pm 1.34$ | $78.07 \pm 1.19$ | +0.17 |
| GPT-3.5 | LLaMA3-8B & Mistral-7B | 59.11 | 77.90 | $80.94 \pm 1.51$ | $80.61 \pm 1.72$ | +2.71 |
| LLaMA3-70B | LLaMA3-8B | 70.16 | 81.77 | $84.97 \pm 1.73$ | $85.69 \pm 1.49$ | +3.92 |
| LLaMA3-70B | Mistral-7B | 70.16 | 81.77 | $82.65 \pm 1.82$ | $84.03 \pm 1.47$ | +2.26 |
| LLaMA3-70B | LLaMA3-8B & Mistral-7B | 70.16 | 81.77 | $84.53 \pm 1.26$ | $84.48 \pm 1.36$ | +2.71 |
| LLaMA3-8B | LLaMA3-8B | 45.85 | 64.64 | $70.83 \pm 1.91$ | $72.93 \pm 2.15$ | +8.29 |
| Mistral-7B | LLaMA3-8B | 19.88 | 40.88 | $40.55 \pm 1.82$ | $42.43 \pm 1.78$ | +1.55 |

Table A7: Results on 70 MATH problems with 16 TR-TA pairs.

| Teacher | Student | SC | Self-Eval | **M1** (MAX) | **M1** (SUM) |
|---|---|---|---|---|---|
| GPT-4 | GPT-4 | 67.14 | 68.57 | 70.00 | 72.86 |
| GPT-4 | GPT-3.5 | 67.14 | 68.57 | 71.43 | 72.86 |
| GPT-3.5 | GPT-3.5 | 52.86 | 52.86 | 57.14 | 58.57 |
| GPT-3.5 | Mistral-7B | 52.86 | 52.86 | 54.29 | 57.14 |

## A.3 Competition-Level Code Synthesis

### A.3.1 Prompt Design

---

**Prompt 3. Teacher Prompt (Coding)**

**[User:]**
[[Question]]:

{TP}

First, let's think step by step to find a complete problem-solving strategy.
Then, write a Python code based on the problem-solving strategy.

**[Assistant:]**
[[RATIONALE]]:

---

**Prompt 4. Student Prompt (Coding)**

**[User:]**
[[Question]]:

Here is an example question, please understand it very carefully:

{TP}

First, let's think step by step to find a complete problem-solving strategy.
Then, write a Python code based on the problem-solving strategy.

---

[[RATIONALE]]:

{TR}

[[Final Code]]:

{TA}

[[Question]]:

Please first understand the problem-solving approach in the rationale of the aforementioned example, and then follow the example to solve the following similar type of problem:

{EP}

First, let's think step by step to find a complete problem-solving strategy.
Then, write a Python code based on the problem-solving strategy.

**[Assistant:]**
[[RATIONALE]]:

---

### Prompt 5. Self-Debugging Prompt (Coding)

**[User:]**
[[Question]]:

{TP or EP}

[[RATIONALE]]:

{TR or ER}

[[Final Code]]:

{TA or EA}

You need to debug this code with the following rules:

(1) If you think the provided code is correct, you must retrieve the original correct code.

(2) If you think the provided code is incorrect, you debug the code and write the final bug-free code.

(3) If there is no complete code, you must write a complete code based on the rationale.

Let's think step by step and remember you **must** give me a complete Python code finally.

**[Assistant:]**

## A.3.2 Additional Experimental Setups

To get the problems from LeetCode, we follow the approach of Reflexion [67] to use LeetCode's official API to obtain all datasets from the Grandmaster dynamic programming study plan. We also employ GPT-3.5 to extract all visible examples for offline evaluation. Additionally, we use the LeetCode Python API to submit the Python code and evaluate the code on invisible examples.

As shown in Alg. A1, within the for loop, we first have the teacher LLM sample TR and TA. To validate LbT, we introduce randomness at this step to generate a variety of TR-TA pairs. For GPT-3.5-0613, we set the temperature and top-P to 1, while for the LLaMA3 family, we set the temperature to 0.6 and top-P to 0.9 (default setting). Next, we prompt the student LLM with the sampled TR-TA as ICL examples to solve EPs. In this step, we use greedy sampling.

Table A9: **S-score** results with standard deviation on *Game Theory* dataset in LeetCode Grandmaster DP study plan. "SG-1"-"SG-4" and "PW" are abbreviations of individual questions in the *Game Theory* dataset. The standard deviations are calculated using the bootstrap sampling technique [43], where size-4 subsets are sampled from the 8 TR-TA pairs as the TR-TA set for **M1**, and the standard deviation of performance is computed across 20 sets.

| Models | Metrics | SG-1 | SG-2 | SG-3 | SG-4 | PW |
|---|---|---|---|---|---|---|
| T=LLaMA3-8B
S=LLaMA3-8B | Avg. | $0.198 \pm 0.125$ | $0.004 \pm 0.002$ | $0.209 \pm 0.068$ | $0.564 \pm 0.084$ | $0.613 \pm 0.128$ |
|  | **M1** (MAX) | $0.539 \pm 0.162$ | $0.004 \pm 0.004$ | $0.220 \pm 0.116$ | $0.690 \pm 0.233$ | $0.576 \pm 0.346$ |
|  | Avg. (V-score=1) | $1.000 \pm 0.000$ | - | - | $0.647 \pm 0.294$ | $0.847 \pm 0.113$ |
|  | **M1** (MAX) (V-score=1) | $1.000 \pm 0.000$ | - | - | $0.724 \pm 0.332$ | $0.939 \pm 0.124$ |
| T=LLaMA3-8B
S=LLaMA3-8B
(w. Self-Debugging) | Avg. | $0.335 \pm 0.142$ | $0.005 \pm 0.002$ | $0.292 \pm 0.108$ | $0.591 \pm 0.106$ | $0.695 \pm 0.076$ |
|  | **M1** (MAX) | $0.382 \pm 0.242$ | $0.009 \pm 0.004$ | $0.503 \pm 0.130$ | $0.728 \pm 0.154$ | $0.723 \pm 0.121$ |
|  | Avg. (V-score=1) | $0.827 \pm 0.147$ | - | - | $0.653 \pm 0.318$ | $0.890 \pm 0.104$ |
|  | **M1** (MAX) (V-score=1) | $0.928 \pm 0.196$ | - | - | $0.815 \pm 0.346$ | $0.941 \pm 0.072$ |
| T=GPT-3.5
S=GPT-3.5 | Avg. | $0.582 \pm 0.128$ | $0.007 \pm 0.002$ | $0.432 \pm 0.177$ | $1.000 \pm 0.000$ | $0.643 \pm 0.132$ |
|  | **M1** (MAX) | $0.827 \pm 0.178$ | $0.010 \pm 0.002$ | $0.631 \pm 0.193$ | $1.000 \pm 0.000$ | $0.774 \pm 0.129$ |
|  | Avg. (V-score=1) | $0.993 \pm 0.006$ | - | $0.746 \pm 0.299$ | $1.000 \pm 0.000$ | $0.914 \pm 0.082$ |
|  | **M1** (MAX) (V-score=1) | $1.000 \pm 0.000$ | - | $0.593 \pm 0.432$ | $1.000 \pm 0.000$ | $0.962 \pm 0.049$ |
| T=GPT-3.5
S=GPT-3.5
(w. Self-Debugging) | Avg. | $0.723 \pm 0.162$ | $0.096 \pm 0.119$ | $0.586 \pm 0.136$ | $1.000 \pm 0.000$ | $0.841 \pm 0.070$ |
|  | **M1** (MAX) | $1.000 \pm 0.000$ | $0.255 \pm 0.368$ | $0.655 \pm 0.323$ | $1.000 \pm 0.000$ | $0.911 \pm 0.104$ |
|  | Avg. (V-score=1) | $0.996 \pm 0.004$ | $1.000 \pm 0.000$ | $0.666 \pm 0.338$ | $1.000 \pm 0.000$ | $0.897 \pm 0.088$ |
|  | **M1** (MAX) (V-score=1) | $1.000 \pm 0.000$ | $1.000 \pm 0.000$ | $0.666 \pm 0.338$ | $1.000 \pm 0.000$ | $0.931 \pm 0.075$ |
| T=LLaMA3-70B
S=LLaMA3-8B | Avg. | $0.838 \pm 0.119$ | $0.008 \pm 0.001$ | $0.677 \pm 0.135$ | $1.000 \pm 0.000$ | $0.597 \pm 0.102$ |
|  | **M1** (MAX) | $0.900 \pm 0.200$ | $0.007 \pm 0.002$ | $0.787 \pm 0.398$ | $1.000 \pm 0.000$ | $0.671 \pm 0.112$ |
|  | Avg. (V-score=1) | $1.000 \pm 0.000$ | - | $1.000 \pm 0.000$ | $1.000 \pm 0.000$ | $0.918 \pm 0.127$ |
|  | **M1** (MAX) (V-score=1) | $1.000 \pm 0.000$ | - | $1.000 \pm 0.000$ | $1.000 \pm 0.000$ | $0.965 \pm 0.112$ |

Table A8: The question IDs and the question title names of problems in *Game Theory*, *Bitmasking*, *General-1D*, and *Tricky Invariant* datasets. Note that only the *Tricky Invariant* dataset is related to the binary search, while the remaining three datasets focus on dynamic programming.

| Dataset | Question ID | Question Title | Question Title Abbr. |
|---|---|---|---|
| Game Theory | 486 | Predict the Winner | PW |
|  | 877 | Stone Game | SG-1 |
|  | 1140 | Stone Game II | SG-2 |
|  | 1406 | Stone Game III | SG-3 |
|  | 1510 | Stone Game IV | SG-4 |
| Bitmasking | 698 | Partition to K Equal Sum Subsets | PKE |
|  | 465 | Optimal Account Balancing | OAB |
|  | 847 | Shortest Path Visiting All Nodes | SPV |
|  | 1125 | Smallest Sufficient Team | SST |
|  | 1434 | Number of Ways to Wear Different Hats to Each Other | NWW |
|  | 1799 | Maximize Score After N Operations | MSA |
| General-1D | 1048 | Longest String Chain | LSC |
|  | 376 | Wiggle Subsequence | WS |
|  | 651 | 4 Keys Keyboard | 4KK |
|  | 32 | Longest Valid Parentheses | LVP |
|  | 1416 | Restore The Array | RTA |
|  | 1259 | Handshakes That Don't Cross | HTD |
|  | 639 | Decode Ways II | DW-2 |
| Tricky Invariant | 1539 | Kth Missing Positive Number | KMPN |
|  | 275 | H Index II | HI2 |

### A.3.3 Additional Results

As shown in Table A9, we use bootstrap sampling (choose 4 TR-TAs from 8 TR-TAs, 20 sets are sampled) to produce the results in Table 3 with standard deviations.

Table A10: Ablation of model settings on *Game Theory* dataset in LeetCode Grandmaster DP study plan: The teacher and studnet belong to different model families. The results of M1 that improve (degrade) by more than 0.01 are highlighted in green (red).

| Models | Metrics | SG-1 | SG-2 | SG-3 | SG-4 | PW |
|---|---|---|---|---|---|---|
| T=LLaMA3-8B S=GPT-3.5 | Avg. | 0.215 | 0.004 | 0.216 | 0.604 | 0.609 |
| | **M1** (MAX) | 0.101 | 0.005 | 0.524 | 0.611 | 0.462 |
| | Avg. (V-score=1) | 1 | - | - | 0.755 | 0.851 |
| | **M1** (MAX) (V-score=1) | 1 | - | - | 1 | 0.871 |
| T=LLaMA3-8B S=GPT-3.5 (w. Self-Debugging) | Avg. | 0.348 | 0.004 | 0.319 | 0.608 | 0.694 |
| | **M1** (MAX) | 0.370 | 0 | 0.565 | 1 | 1 |
| | Avg. (V-score=1) | 0.797 | - | - | 0.722 | 0.851 |
| | **M1** (MAX) (V-score=1) | 0.391 | - | - | 1 | 1 |
| T=GPT-3.5 S=LLaMA3-8B | Avg. | 0.582 | 0.007 | 0.428 | 1 | 0.645 |
| | **M1** (MAX) | 0.652 | 0.011 | 0.681 | 1 | 0.766 |
| | Avg. (V-score=1) | 0.994 | - | 0.712 | 1 | 0.867 |
| | **M1** (MAX) (V-score=1) | 0.989 | - | 0.712 | 1 | 0.766 |
| T=GPT-3.5 S=LLaMA3-8B (w. Self-Debugging) | Avg. | 0.701 | 0.133 | 0.591 | 1 | 0.853 |
| | **M1** (MAX) | 1 | 0.204 | 0.668 | 1 | 0.867 |
| | Avg. (V-score=1) | 0.996 | 1 | 0.712 | 1 | 0.911 |
| | **M1** (MAX) (V-score=1) | 1 | 1 | 1 | 1 | 0.867 |

Table A11: **S-score** results on *Bitmasking* dataset in LeetCode Grandmaster DP study plan. Here, the teacher and student are the same. This dataset is too difficult for LLaMA3-8B, which can hardly solve these coding problems.

| Models | Metrics | PKE | OAB | SPV | SST | NWW | MSA |
|---|---|---|---|---|---|---|---|
| T=LLaMA3-8B S=LLaMA3-8B | Avg. | 0.458 | 0.132 | 0.009 | 0.016 | 0.006 | 0.110 |
| | **M1** (MAX) | 0.458 | 0.278 | 0.005 | 0 | 0.007 | 0.110 |
| | Avg. (V-score=1) | 0.628 | - | - | - | - | - |
| | **M1** (MAX) (V-score=1) | 0.628 | - | - | - | - | - |
| T=LLaMA3-8B S=LLaMA3-8B (w. Self-Debugging) | Avg. | 0.463 | 0.135 | 0.005 | 0.039 | 0.006 | 0.109 |
| | **M1** (MAX) | 0.908 | 0.135 | 0.006 | 0.045 | 0 | 0.108 |
| | Avg. (V-score=1) | 0.642 | 0.472 | - | - | - | - |
| | **M1** (MAX) (V-score=1) | 0.908 | 0.472 | - | - | - | - |
| T=GPT-3.5 S=GPT-3.5 | Avg. | 0.788 | 0.024 | 0.880 | 0.148 | 0.369 | 0.258 |
| | **M1** (MAX) | 0.936 | 0 | 1 | 1 | 0.923 | 0.584 |
| | Avg. (V-score=1) | 0.901 | 0.111 | 1 | 0.526 | 0.949 | 0.584 |
| | **M1** (MAX) (V-score=1) | 0.936 | 0.111 | 1 | 1 | 0.923 | 0.584 |
| T=GPT-3.5 S=GPT-3.5 (w. Self-Debugging) | Avg. | 0.788 | 0.024 | 0.880 | 0.148 | 0.369 | 0.256 |
| | **M1** (MAX) | 0.943 | 0 | 1 | 1 | 1 | 0.481 |
| | Avg. (V-score=1) | 0.901 | 0.111 | 1 | 0.526 | 0.949 | 0.578 |
| | **M1** (MAX) (V-score=1) | 0.943 | 0.111 | 1 | 1 | 1 | 0.578 |

Table A12: **S-score** results on *Bitmasking* dataset in LeetCode Grandmaster DP study plan. Here, the teacher and student are different.

| Models | Metrics | PKE | OAB | SPV | SST | NWW | MSA |
|---|---|---|---|---|---|---|---|
| T=LLaMA3-8B S=GPT-3.5 | Avg. | 0.458 | 0.132 | 0.009 | 0.016 | 0.006 | 0.110 |
| | **M1** (MAX) | 0.551 | 0.014 | 0 | 0 | 0 | 0.162 |
| | Avg. (V-score=1) | 0.628 | - | - | - | - | - |
| | **M1** (MAX) (V-score=1) | 0.488 | - | - | - | - | - |
| T=LLaMA3-8B S=GPT-3.5 (w. Self-Debugging) | Avg. | 0.463 | 0.135 | 0.005 | 0.039 | 0.006 | 0.109 |
| | **M1** (MAX) | 0.537 | 0 | 0 | 0.039 | 0 | 0.130 |
| | Avg. (V-score=1) | 0.642 | 0.472 | - | - | - | - |
| | **M1** (MAX) (V-score=1) | 0.908 | 0.472 | - | - | - | - |
| T=GPT-3.5 S=LLaMA3-8B | Avg. | 0.788 | 0.024 | 0.880 | 0.148 | 0.369 | 0.258 |
| | **M1** (MAX) | 0.788 | 0.039 | 1 | 0.037 | 0 | 0.203 |
| | Avg. (V-score=1) | 0.901 | 0.111 | 1 | 0.526 | 0.949 | 0.584 |
| | **M1** (MAX) (V-score=1) | 0.901 | 0.111 | 1 | 0.053 | 0.949 | 0.584 |
| T=GPT-3.5 S=LLaMA3-8B (w. Self-Debugging) | Avg. | 0.788 | 0.024 | 0.880 | 0.148 | 0.369 | 0.256 |
| | **M1** (MAX) | 0.926 | 0.024 | 1 | 0.197 | 1 | 0.286 |
| | Avg. (V-score=1) | 0.901 | 0.111 | 1 | 0.526 | 0.949 | 0.578 |
| | **M1** (MAX) (V-score=1) | 0.926 | 0.111 | 1 | 0.526 | 1 | 0.578 |

Table A13: **S-score** results on *General-1D* dataset in LeetCode Grandmaster DP study plan. Here, the teacher and student are the same.

| Models | Metrics | LSC | WS | 4KK | LVP | RTA | HTD | DW-2 |
|---|---|---|---|---|---|---|---|---|
| T=LLaMA3-8B S=LLaMA3-8B | Avg. | 0.326 | 0.819 | 0.062 | 0.563 | 0.108 | 0.013 | 0.507 |
| | **M1** (MAX) | 0.244 | 1 | 0.090 | 1 | 0.151 | 0.027 | 0.679 |
| | Avg. (V-score=1) | 1 | 1 | - | 0.671 | - | - | - |
| | **M1** (MAX) (V-score=1) | 1 | 1 | - | 1 | - | - | - |
| T=LLaMA3-8B S=LLaMA3-8B (w. Self-Debugging) | Avg. | 0.273 | 0.399 | 0.085 | 0.567 | 0.106 | 0.013 | 0.507 |
| | **M1** (MAX) | 0.288 | 0.500 | 0.120 | 0.719 | 0.151 | 0.020 | 0.679 |
| | Avg. (V-score=1) | 1 | 1 | - | 0.607 | - | - | - |
| | **M1** (MAX) (V-score=1) | 1 | 1 | - | 0.719 | - | - | - |
| T=GPT-3.5 S=GPT-3.5 | Avg. | 1 | 1 | 0.542 | 0.818 | 0.089 | 0.653 | 0.565 |
| | **M1** (MAX) | 1 | 1 | 1 | 1 | 0.128 | 1 | 0.697 |
| | Avg. (V-score=1) | 1 | 1 | 1 | 1 | 0.128 | 0.867 | 0.719 |
| | **M1** (MAX) (V-score=1) | 1 | 1 | 1 | 1 | 0.128 | 1 | 0.697 |
| T=GPT-3.5 S=GPT-3.5 (w. Self-Debugging) | Avg. | 1 | 1 | 0.547 | 0.818 | 0.089 | 0.653 | 0.549 |
| | **M1** (MAX) | 1 | 1 | 1 | 1 | 0.126 | 1 | 0.587 |
| | Avg. (V-score=1) | 1 | 1 | 1 | 1 | 0.128 | 0.867 | 0.719 |
| | **M1** (MAX) (V-score=1) | 1 | 1 | 1 | 1 | 0.198 | 1 | 0.697 |

Table A14: **S-score** results on *General-1D* dataset in LeetCode Grandmaster DP study plan. Here, the teacher and student are different.

| Models | Metrics | LSC | WS | 4KK | LVP | RTA | HTD | DW-2 |
|---|---|---|---|---|---|---|---|---|
| T=LLaMA3-8B S=GPT-3.5 | Avg. | 0.326 | 0.819 | 0.062 | 0.563 | 0.108 | 0.013 | 0.507 |
| | **M1** (MAX) | 0.253 | 0.774 | 0.080 | 0.573 | 0.081 | 0.007 | 0.342 |
| | Avg. (V-score=1) | 1 | 1 | - | 0.671 | - | - | - |
| | **M1** (MAX) (V-score=1) | 1 | 1 | - | 0.719 | - | - | - |
| T=LLaMA3-8B S=GPT-3.5 (w. Self-Debugging) | Avg. | 0.273 | 0.399 | 0.073 | 0.567 | 0.106 | 0.013 | 0.507 |
| | **M1** (MAX) | 0.485 | 0.661 | 0.030 | 1 | 0.151 | 0.016 | 0.561 |
| | Avg. (V-score=1) | 1 | 1 | - | 0.607 | - | - | - |
| | **M1** (MAX) (V-score=1) | 1 | 1 | - | 1 | - | - | - |
| T=GPT-3.5 S=LLaMA3-8B | Avg. | 1 | 1 | 0.542 | 0.818 | 0.089 | 0.653 | 0.565 |
| | **M1** (MAX) | 1 | 1 | 0.520 | 1 | 0.058 | 0.013 | 0.697 |
| | Avg. (V-score=1) | 1 | 1 | 1 | 1 | 0.128 | 0.867 | 0.719 |
| | **M1** (MAX) (V-score=1) | 1 | 1 | 1 | 1 | 0.128 | 0.867 | 0.697 |
| T=GPT-3.5 S=LLaMA3-8B (w. Self-Debugging) | Avg. | 1 | 1 | 0.547 | 0.818 | 0.089 | 0.653 | 0.549 |
| | **M1** (MAX) | 1 | 1 | 0.560 | 1 | 0.052 | 1 | 0.697 |
| | Avg. (V-score=1) | 1 | 1 | 1 | 1 | 0.128 | 0.867 | 0.719 |
| | **M1** (MAX) (V-score=1) | 1 | 1 | 1 | 1 | 0.128 | 1 | 0.697 |

### A.3.4 Examples

Student models can make both logical and non-logical errors. The non-logical errors – such as missing imports, miswritten variable names, improper handling of simple boundary conditions, or incorrect usage of library functions – are mainly related to the knowledge required by specific EPs and the robustness of the student model, rather than the quality of the TR-TA pair. Thus, these errors do not provide helpful feedback to the teacher. To address this, we apply self-debugging to correct non-logical bugs, making the students' exam V-score more reflective of the TR-TA quality and therefore more helpful to the teacher. We show an example of how self-debugging resolves a simple boundary condition bug in Ex. 6.

Regarding logical errors, our experiments show that students generally follow the TR-TA pair in making logical errors or writing inefficient logic. In such cases, the students' exam V-score provides valuable feedback for the teacher. We have discussed the examples Exs. 1 to 4 in § 3.3.2. Nevertheless, there exist cases where the student fails to follow the logic of the TR-TA pair to solve the EPs. We find that this undesirable situation is more likely to occur when the student and teacher come from different model families. An example of a failure case is shown in Ex. 5. This discrepancy is highlighted by the comparisons between Tabs. 3 and A10, between Tabs. A11 and A12, and between Tabs. A13 and A14, where negative cases are more frequent in Tabs. A10, A12 and A14 when the teacher and student belong to different model families.

---

**Example 1. M1 on Code Synthesis 1 (Teacher=Student=GPT-3.5-turbo-0613)**

**TP:**
**(Leetcode Question ID: 877)** Write a python code """class Solution(): def stoneGame(self, piles: List[int]) -> bool:""" to solve the following problem:
Alice and Bob play a game with piles of stones. There are an even number of piles arranged in a row,
...

- - - - - - - - - - - - - - - - - - - - - - - - - - - - - - - - - - - - - - - - - - - - - - - -

**TR and TA:**
To determine the winner of the game, Alice or Bob, you can use a dynamic programming approach. Let's think step by step:

1. Create a 2D array: Create a 2D array of size 'n' by 'n', where 'n' is the length of the 'piles' list. This array will store the maximum number of stones a player can collect from piles 'i' to 'j'.

2. Base case: Initialize the diagonal elements of the 2D array with the values of the piles themselves, as Alice and Bob can only choose from one pile each.

3. Dynamic programming iteration: Iterate over the 2D array diagonally, dynamically computing the maximum number of stones a player can collect from piles 'i' to 'j'. The maximum number of stones a player can collect is the maximum of two options:
a) Taking the pile at index 'i' and the maximum number of stones from piles 'i+1' to 'j' (Alice's turn), or

b) Taking the pile at index 'j' and the maximum number of stones from piles 'i' to 'j-1' (Bob's turn).

4. Return the winner: After the dynamic programming iteration, check if the maximum number of stones Alice can collect (from piles '[0, n-1]') is greater than the maximum number of stones Bob can collect. If so, return True; otherwise, return False.

```
1    from typing import List
2
3    class Solution:
4      def stoneGame(self, piles: List[int]) -> bool:
5        n = len(piles)
6        dp = [[0] * n for _ in range(n)]
7
8        # Base case: diagonal elements
9        for i in range(n):
10           dp[i][i] = piles[i]
11
12       # Dynamic programming iteration
13       for length in range(2, n+1):
14           for i in range(n - length + 1):
15               j = i + length - 1
16               dp[i][j] = max(piles[i] - dp[i+1][j], piles[j] - dp[i][j-1])
17
18       # Return the winner
19       return dp[0][n-1] > 0
```

**EP:**
(Leetcode Question ID: 486) Write a python code """class Solution(): def predictTheWinner(self, nums: List[int]) -> bool:""" to solve the following problem:
You are given an integer array 'nums'. Two players are playing a game with this array: player 1 and player 2.
...

**ER and EA:**
To determine if Player 1 can win the game, you can use a dynamic programming approach. Let's think step by step:

1. Create a 2D array: Create a 2D array of size 'n' by 'n', where 'n' is the length of the 'nums' list. This array will store the maximum score difference between Player 1 and Player 2 when considering the subarray from index 'i' to index 'j'.

2. Base case: Initialize the diagonal elements of the 2D array with the values of the numbers in the 'nums' list, as Player 1 and Player 2 can only choose one number each.

3. Dynamic programming iteration: Iterate over the 2D array diagonally, dynamically computing the maximum score difference between Player 1 and Player 2 when considering the subarray from index 'i' to index 'j'. The maximum score difference is the maximum of two options:
a) The number at index 'i' minus the maximum score difference between Player 1 and Player 2 when considering the subarray from index 'i+1' to index 'j' (Player 1's turn), or
b) The number at index 'j' minus the maximum score difference between Player 1 and Player 2 when considering the subarray from index 'i' to index 'j-1' (Player 2's turn).

4. Return the result: After the dynamic programming iteration, check if the maximum score difference between Player 1 and Player 2 when considering the entire array (from index '0' to index 'n-1') is greater than or equal to '0'. If so, return True; otherwise, return False.

```
1    from typing import List
2
3    class Solution:
4        def predictTheWinner(self, nums: List[int]) -> bool:
5            n = len(nums)
6            dp = [[0] * n for _ in range(n)]
7
8            # Base case: diagonal elements
9            for i in range(n):
10               dp[i][i] = nums[i]
11
12           # Dynamic programming iteration
13           for length in range(2, n+1):
14               for i in range(n - length + 1):
15                   j = i + length - 1
16                   dp[i][j] = max(nums[i] - dp[i+1][j], nums[j] - dp[i][j-1])
17
18           # Return the result
19           return dp[0][n-1] >= 0
```

**TP:**
(Leetcode Question ID: 877) Write a python code """class Solution(): def stoneGame(self, piles: List[int]) -> bool:""" to solve the following problem:
Alice and Bob play a game with piles of stones. There are an even number of piles arranged in a row,
...

- - - - - - - - - - - - - - - - - - - - - - - - - - - - - - - - - - - - - - - - - - - - - - - - -

**TR and TA:**
To solve this problem, we can use a dynamic programming approach. Let's break down the problem-solving strategy step by step:

1. Initialize Variables: The code initializes two variables 'n' and 'm' to keep track of the total number of piles and an empty cache dictionary.

2. Recursive Function: The code defines a recursive function called 'aliceWins' which takes two parameters: 'start' and 'end'. This function represents the state of the game - the range of piles currently available.

3. Base Case: The base case of the recursive function is when there are no more piles left, i.e., when 'start > end'. In this case, we return True if Alice's total number of stones is greater than Bob's total number of stones, otherwise return False.

4. Memoization: Before calculating the result for the current state, we check if the result is already present in the cache dictionary. If so, we return the cached result instead of recalculating it.

5. Maximum Stones Calculation: We calculate the maximum number of stones Alice can obtain if she starts from the 'start' pile and Bob plays optimally. We have two options:
- Alice takes the 'start' pile: In this case, Alice's total number of stones will be the 'start' pile minus the maximum number of stones Bob can obtain from the remaining piles. Bob's maximum number of stones can be obtained using the 'aliceWins' function recursively by changing the 'start' index to 'start+1' and keeping the 'end' index the same.
- Alice takes the 'end' pile: In this case, Alice's total number of stones will be the 'end' pile minus the maximum number of stones Bob can obtain from the remaining piles. Bob's maximum number of stones can be obtained using the 'aliceWins' function recursively by changing the 'end' index to 'end-1' and keeping the 'start' index the same.

6. Update Cache and Return Result: After calculating the maximum number of stones Alice can obtain, we update the cache dictionary with the result for the current state '(start, end)' and return the result.

7. Main Function: The code defines the main 'stoneGame' function that calls the 'aliceWins' function with the initial state '(0, n-1)' and returns the result.

```
1   from typing import List
2
3   class Solution():
4       def stoneGame(self, piles: List[int]) -> bool:
5           n = len(piles)
6           cache = {}  # Cache dictionary to store results
7
8           def aliceWins(start: int, end: int) -> bool:
9               # Base case - no more piles left
10              if start > end:
11                  return sum(piles) % 2 == 1
12
13              # Check if result is already cached
14              if (start, end) in cache:
15                  return cache[(start, end)]
16
17              # Option 1: Alice takes the start pile
18              option1 = piles[start] - aliceWins(start + 1, end)
19
20              # Option 2: Alice takes the end pile
21              option2 = piles[end] - aliceWins(start, end - 1)
22
23              # Update cache with the result and return
24              cache[(start, end)] = max(option1, option2) > 0
25              return cache[(start, end)]
26
27          # Call the recursive function with initial state (0, n-1)
28          return aliceWins(0, n-1)
```

- - - - - - - - - - - - - - - - - - - - - - - - - - - - - - - - - - - - - - - - - - - - - - - - -

**EP:**
(Leetcode Question ID: 486) Write a python code """class Solution(): def predictTheWinner(self, nums: List[int]) -> bool:""" to solve the following problem:
You are given an integer array 'nums'. Two players are playing a game with this array: player 1 and player 2.
...

- - - - - - - - - - - - - - - - - - - - - - - - - - - - - - - - - - - - - - - - - - - - - - - - -

**ER and EA:**

To solve this problem, we can use a dynamic programming approach. Let's break down the problem-solving strategy step by step:

1. Initialize Variables: The code initializes two variables 'n' and 'm' to keep track of the total number of elements in the array and an empty cache dictionary.

2. Recursive Function: The code defines a recursive function called 'player1Wins' which takes two parameters: 'start' and 'end'. This function represents the state of the game - the range of elements currently available.

3. Base Case: The base case of the recursive function is when there are no more elements left, i.e., when 'start > end'. In this case, we return True if Player 1's score is greater than or equal to Player 2's score, otherwise return False.

4. Memoization: Before calculating the result for the current state, we check if the result is already present in the cache dictionary. If so, we return the cached result instead of recalculating it.

5. Maximum Score Calculation: We calculate the maximum score Player 1 can obtain if they start from the 'start' element and both players play optimally. We have two options:
- Player 1 takes the 'start' element: In this case, Player 1's score will be the 'start' element plus the minimum score Player 2 can obtain from the remaining elements. Player 2's minimum score can be obtained using the 'player1Wins' function recursively by changing the 'start' index to 'start+1' and keeping the 'end' index the same.
- Player 1 takes the 'end' element: In this case, Player 1's score will be the 'end' element plus the minimum score Player 2 can obtain from the remaining elements. Player 2's minimum score can be obtained using the 'player1Wins' function recursively by changing the 'end' index to 'end-1' and keeping the 'start' index the same.

6. Update Cache and Return Result: After calculating the maximum score Player 1 can obtain, we update the cache dictionary with the result for the current state '(start, end)' and return the result.

7. Main Function: The code defines the main 'predictTheWinner' function that calls the 'player1Wins' function with the initial state '(0, n-1)' and returns the result.

```python
from typing import List

class Solution():
    def predictTheWinner(self, nums: List[int]) -> bool:
        n = len(nums)
        cache = {}  # Cache dictionary to store results

        def player1Wins(start: int, end: int) -> bool:
            # Base case - no more elements left
            if start > end:
                return True

            # Check if result is already cached
            if (start, end) in cache:
                return cache[(start, end)]

            # Option 1: Player 1 takes the start element
            option1 = nums[start] + min(player1Wins(start + 1, end),
                player1Wins(start, end - 1))

            # Option 2: Player 1 takes the end element
            option2 = nums[end] + min(player1Wins(start + 1, end),
                player1Wins(start, end - 1))

            # Update cache with the result and return
            cache[(start, end)] = option1 >= option2
            return cache[(start, end)]

        # Call the recursive function with initial state (0, n-1)
        return player1Wins(0, n-1)
```

---

### Example 3. **M1 on Code Synthesis 3 (Teacher=Student=GPT-3.5-turbo-0613)**

**TP:**

**(Leetcode Question ID: 877)** Write a python code """class Solution(): def stoneGame(self, piles: List[int]) -> bool:""" to solve the following problem:
Alice and Bob play a game with piles of stones. There are an even number of piles arranged in a row,
...

- - - - - - - - - - - - - - - - - - - - - - - - - - - - - - - - - - - - - - - - - - - - - - - - - - - -

**TR and TA:**

The problem is asking to determine if Alice can win the game given a list of piles of stones. Here's a step-by-step strategy to solve the problem:

1. Determine Alice's optimal strategy:
- Since Alice starts first, she has the advantage of making the first move.
- Alice's goal is to end with the most stones, so she will try to maximize her score.
- In each turn, Alice can choose to take the entire pile from either the beginning or the end of the row.

2. Determine the total number of stones:
- The total number of stones is odd, which means there will be no ties.
- Alice can always take the largest pile in each turn.
- Calculate the total number of stones by summing all the elements in the piles list.

3. Check if Alice can always win:
- If Alice takes all the piles from either the beginning or the end in each turn, she will always end up with more stones than Bob.
- Check if the total number of stones is greater than half of the total stones.

4. Return the result:
- If Alice can always win, return True. Otherwise, return False.

```
1   from typing import List
2
3   class Solution():
4       def stoneGame(self, piles: List[int]) -> bool:
5           total_stones = sum(piles)
6           half_total = total_stones // 2
7
8           if half_total <= max(piles):
9               return True
10          else:
11              return False
```

**EP:**

(**Leetcode Question ID: 486**) Write a python code """class Solution(): def predictTheWinner(self, nums: List[int]) -> bool:""" to solve the following problem:
You are given an integer array 'nums'. Two players are playing a game with this array: player 1 and player 2.
...

**ER and EA:**

The problem is asking to determine if Player 1 can win the game given an array of numbers. Here's a step-by-step strategy to solve the problem:

1. Determine Player 1's optimal strategy:
- Player 1 starts first and wants to maximize their score.
- In each turn, Player 1 can choose a number from either end of the array.

2. Determine the total number of elements:
- Calculate the total number of elements in the array.

3. Check if Player 1 can always win:
- If the total number of elements is odd, Player 1 can always win by choosing the numbers alternately.
- If the total number of elements is even, Player 1 can still win if their score is greater than or equal to Player 2's score.

4. Return the result:
- If Player 1 can always win or has a higher score than Player 2, return True. Otherwise, return False.

```
1   from typing import List
2
3   class Solution():
4       def predictTheWinner(self, nums: List[int]) -> bool:
5           total_elements = len(nums)
6
7           if total_elements % 2 == 1:
8               return True
9           else:
10              return sum(nums) >= sum(nums) // 2
```

Example 4. **M1 on Code Synthesis 4 (Teacher=Student=GPT-3.5-turbo-0613)**

**TP:**

(**Leetcode Question ID: 1406**) Write a python code """class Solution(): def stoneGameIII(self, stoneValue: List[int]) -> str:""" to solve the following problem:

Alice and Bob continue their games with piles of stones. There are several stones arranged in a row, and each stone has an associated value which is an integer given in the array 'stoneValue'.
...

--------------------------------------------------------------------------------

**TR and TA:**

To solve this problem, we can use dynamic programming to determine the optimal strategy for each player.
Let's think step by step:

1. Create a helper function called 'dfs', which takes in the current position 'i' and the list of stone values. This function will recursively calculate the maximum score that can be achieved starting from position 'i' and returning the difference of Alice's score and Bob's score.

2. In the 'dfs' function, we need a base case for the end of the game. If the current position 'i' is greater than or equal to the length of the stone values list, we've reached the end of the game, and the difference in scores is 0.

3. If the base case is not met, we need to calculate the maximum score difference achievable starting from the current position. We can iterate through the next 3 positions, taking 1, 2, or 3 stones each time, and recursively calculate the score difference for the opponent. We take the maximum score difference among the opponent's choices and subtract the current stone value to get the score difference for the current player.

4. We return the maximum score difference calculated from the previous step. If the current player is Alice (starting player), a positive score difference means Alice wins. If the current player is Bob, a negative score difference means Bob wins. If the score difference is 0, it's a tie.

5. Finally, in the 'stoneGameIII' function, we can call the 'dfs' function with the initial position 0 and the stone values list. We compare the result of 'dfs' with 0 to determine the winner.

```
1   from typing import List
2
3   class Solution():
4       def stoneGameIII(self, stoneValue: List[int]) -> str:
5           def dfs(i: int, stoneValues: List[int]) -> int:
6               if i >= len(stoneValues):
7                   return 0
8
9               maxScore = float('-inf')
10              for j in range(i, min(i+3, len(stoneValues))):
11                  currScore = sum(stoneValues[i:j+1])
12                  opponentScoreDiff = dfs(j + 1, stoneValues)
13                  maxScore = max(maxScore, currScore - opponentScoreDiff)
14
15              return maxScore
16
17          scoreDiff = dfs(0, stoneValue)
18          if scoreDiff > 0:
19              return 'Alice'
20          elif scoreDiff < 0:
21              return 'Bob'
22          else:
23              return 'Tie'
```

--------------------------------------------------------------------------------

**EP:**

(Leetcode Question ID: 486) Write a python code """class Solution(): def predictTheWinner(self, nums: List[int]) -> bool:""" to solve the following problem:
You are given an integer array 'nums'. Two players are playing a game with this array: player 1 and player 2.
...

--------------------------------------------------------------------------------

**ER and EA:**

To solve this problem, we can use dynamic programming to determine the optimal strategy for each player.
Let's think step by step:

1. Create a helper function called 'dfs', which takes in the current start and end indices of the array. This function will recursively calculate the maximum score that can be achieved starting from the current indices and returning the difference of Player 1's score and Player 2's score.

2. In the 'dfs' function, we need a base case for the end of the game. If the start index is greater than the end index, we've reached the end of the game, and the difference in scores is 0.

3. If the base case is not met, we need to calculate the maximum score difference achievable starting from the current indices. We have two options: either Player 1 chooses the number at the start index or Player 1 chooses the number at the end index. We recursively calculate the score difference for the opponent for both options and take the maximum score difference among the opponent's choices. We subtract the opponent's score difference from the current player's chosen number to get the score difference for the current player.

4. We return the maximum score difference calculated from the previous step. If the score difference is positive or equal to 0, it means Player 1 wins.

5. Finally, in the 'predictTheWinner' function, we can call the 'dfs' function with the initial start and end indices (0 and len(nums)-1) of the array. We compare the result of 'dfs' with 0 to determine if Player 1 can win.

```python
from typing import List

class Solution():
    def predictTheWinner(self, nums: List[int]) -> bool:
        def dfs(start: int, end: int) -> int:
            if start > end:
                return 0

            chooseStart = nums[start] - dfs(start + 1, end)
            chooseEnd = nums[end] - dfs(start, end - 1)

            return max(chooseStart, chooseEnd)

        scoreDiff = dfs(0, len(nums) - 1)
        return scoreDiff >= 0
```

---

## Example 5. M1 on Code Synthesis 5 (Teacher=GPT-3.5-turbo-0613 Student=LLaMA3-8B)

**TP:**

(Leetcode Question ID: 486) Write a python code """class Solution(): def predictTheWinner(self, nums: List[int]) -> bool:""" to solve the following problem:
You are given an integer array 'nums'. Two players are playing a game with this array: player 1 and player 2.
...

- - - - - - - - - - - - - - - - - - - - - - - - - - - - - - - - - - - - - - - - - - - - - - - - - - - - - - - - - -

**TR and TA:**

To solve this problem, we can use the concept of dynamic programming. Let's think step by step:

1. Define a function to determine the winner:
- The function should take the array of integers 'nums' as input and return a boolean value indicating whether Player 1 can win the game.
- We will create a 2D memoization table, 'dp', to store the scores difference between Player 1 and Player 2 at each subgame. 'dp[i][j]' represents the maximum score difference when considering the subarray 'nums[i:j+1]'.
- The function will call a helper function 'calculateScores' to calculate the maximum scores at each subgame.

2. Define the 'calculateScores' function:
- The 'calculateScores' function will take the start and end indices, 'start' and 'end', as input and return the maximum score difference between Player 1 and Player 2 for the subarray 'nums[start:end+1]'.
- We will use a recursive approach to consider all possible moves that Player 1 can make.
- Within the recursive function, we will check if the score difference for the current subarray has already been calculated and stored in the memoization table 'dp'. If so, we will return the stored value.
- If the start index is greater than the end index, it means that all elements in the subarray have been chosen, and we will return a score difference of 0.
- Otherwise, we will consider two options:
- If Player 1 chooses the first element ('nums[start]'), the score difference will be the value of 'nums[start]' minus the maximum score difference that Player 2 can obtain in the remaining subarray, calculated recursively.
- If Player 1 chooses the last element ('nums[end]'), the score difference will be the value of 'nums[end]' minus the maximum score difference that Player 2 can obtain in the remaining subarray, calculated recursively.
- We will take the maximum of these two options and store the result in the memoization table 'dp'.
- Finally, we will return the calculated score difference.

3. In the main function 'predictTheWinner', we will initialize the memoization table 'dp' with zeros and call the helper function 'calculateScores' to determine the maximum score difference between Player 1 and Player 2 for the entire array 'nums'.
- If the score difference is greater than or equal to 0, it means that Player 1 can win the game, and we will return 'True'.
- Otherwise, we will return 'False'.

```python
from typing import List

class Solution():
    def predictTheWinner(self, nums: List[int]) -> bool:
        n = len(nums)
        dp = [[0] * n for _ in range(n)]

        def calculateScores(start: int, end: int) -> int:
            if start > end:
                return 0
            if dp[start][end] != 0:
```

```
12                   return dp[start][end]
13
14           chooseStart = nums[start] - calculateScores(start + 1, end)
15           chooseEnd = nums[end] - calculateScores(start, end - 1)
16
17           dp[start][end] = max(chooseStart, chooseEnd)
18
19           return dp[start][end]
20
21       return calculateScores(0, n - 1) >= 0
```

**EP:**

 Write a python code """class Solution(): def winnerSquareGame(self, n: int) -> bool:"""
to solve the following problem:
Alice and Bob take turns playing a game, with Alice starting first.
...

**ER and EA:**

To solve this problem, we can use a similar approach as before. Let's think step by step:

1. Define a function to determine the winner:
- The function should take the integer 'n' as input and return a boolean value indicating whether Alice wins the game.
- We will create a set 'squares' to store all the square numbers less than or equal to 'n'.
- The function will call a helper function 'canWin' to determine if Alice can win the game.

2. Define the 'canWin' function:
- The 'canWin' function will take the current number of stones 'n' as input and return a boolean value indicating whether
Alice can win the game.
- We will use a recursive approach to consider all possible moves that Alice can make.
- Within the recursive function, we will check if the current number of stones is less than or equal to 0. If so, it means that
Alice has lost the game, and we will return 'False'.
- Otherwise, we will consider all the square numbers in the set 'squares' that are less than or equal to 'n'. For each square
number, we will recursively call the 'canWin' function with the updated number of stones.
- If any of the recursive calls return 'True', it means that Alice can win the game, and we will return 'True'.
- If none of the recursive calls return 'True', it means that Alice cannot win the game, and we will return 'False'.

3. In the main function 'winnerSquareGame', we will create the set 'squares' and call the helper function 'canWin' to
determine if Alice can win the game.
- If the result is 'True', we will return 'True'.
- Otherwise, we will return 'False'.

```
1   class Solution():
2       def winnerSquareGame(self, n: int) -> bool:
3           squares = set(i * i for i in range(int(math.sqrt(n)) + 1))
4           memo = {}
5
6           def canWin(n: int) -> bool:
7               if n <= 0:
8                   return False
9               if n in memo:
10                  return memo[n]
11              for square in squares:
12                  if square > n:
13                      break
14                  if not canWin(n - square):
15                      memo[n] = True
16                      return True
17              memo[n] = False
18              return False
19
20          return canWin(n)
```

---

**Example 6. Code Synthesis with Self-Debugging (Debugger=GPT-3.5-turbo-0613)**

**Problem:**

 Write a python code """class Solution(): def stoneGameIII(self, stoneValue: List[int]) ->
str:""" to solve the following problem:
Alice and Bob continue their games with piles of stones. There are several stones arranged in a row, and each stone has an
associated value which is an integer given in the array 'stoneValue'.
...

**Rationale and Answer:**

To solve this problem, we can use dynamic programming. Let's think step by step:

1. Initialize the scores array: We initialize an array 'scores' with a length of 'len(stoneValue) + 1'. The 'scores' array will store the maximum score that can be obtained from each starting index.

2. Iterate from the end: We iterate backwards through the 'stoneValue' array, starting from the last element. For each index 'i', we calculate the maximum score that can be obtained from that index by considering all possible moves: taking '1', '2', or '3' stones.

3. Determine the maximum score: For each possible move, we consider the sum of the stones taken and subtract the maximum score that can be obtained from the next index ('scores[i+1]', 'scores[i+2]', and 'scores[i+3]'). We take the maximum of these values as the maximum score from index 'i' and store it in the 'scores' array.

4. Determine the winner: After calculating the maximum scores for all indices, we compare 'scores[0]' with the sum of all the stone values ('sum(stoneValue)') to determine the winner. If 'scores[0]' is greater, Alice wins. If 'scores[0]' is smaller, Bob wins. Otherwise, it's a tie.

5. Return the result: We return the appropriate string based on the winner determination.

```python
from typing import List

class Solution():
    def stoneGameIII(self, stoneValue: List[int]) -> str:
        n = len(stoneValue)
        scores = [0] * (n + 1)

        for i in range(n-1, -1, -1):
            take1 = stoneValue[i] - scores[i+1]
            take2 = sum(stoneValue[i:i+2]) - scores[i+2]
            take3 = sum(stoneValue[i:i+3]) - scores[i+3]
            scores[i] = max(take1, take2, take3)

        if scores[0] > sum(stoneValue) - scores[0]:
            return "Alice"
        elif scores[0] < sum(stoneValue) - scores[0]:
            return "Bob"
        else:
            return "Tie"
```

- - - - - - - - - - - - - - - - - - - - - - - - - - - - - - - - - - - - - - - - - - - - - - - - - - - -

**Debug Output:**

Based on the provided rationale, here is the final bug-free code:

```python
from typing import List

class Solution():
    def stoneGameIII(self, stoneValue: List[int]) -> str:
        n = len(stoneValue)
        scores = [0] * (n + 1)

        for i in range(n-1, -1, -1):
            take1 = stoneValue[i] - scores[i+1]
            take2 = sum(stoneValue[i:i+2]) - scores[i+2] if i+2 <= n else
                float('-inf')
            take3 = sum(stoneValue[i:i+3]) - scores[i+3] if i+3 <= n else
                float('-inf')
            scores[i] = max(take1, take2, take3)

        if scores[0] > sum(stoneValue) - scores[0]:
            return "Alice"
        elif scores[0] < sum(stoneValue) - scores[0]:
            return "Bob"
        else:
            return "Tie"
```

This code fixes the bug where the index was going out of range when calculating 'take2' and 'take3' by adding conditions to check if 'i+2' and 'i+3' are within the range of the 'stoneValue' list. If they are not, we set the value to 'float('-inf')' to ensure that it is not considered in the maximum calculation.

### A.3.5 Natural Language Rationale has Positive Effect on Code Following in ICL

In the code synthesis task, the answer (TA and EA) refers to the resulting code. We find that **having models produce a natural language rationale (TR and ER) before writing the code is crucial for**

**ICL and ultimately the LbT-based scoring to function as expected**. Specifically, this approach allows the student's output code EA to align more closely with the ICL example code TA, thus the exam score (i.e., LbT score) can better reflect the quality of TA.

As shown in Ex. 8, when we only use TP+TA as the ICL example to teach the student, the EA does not follow the DP coding style in TA and instead employs a recursive method to solve EP. This aligns with the finding in [50] that labels in the ICL examples have limited effect on the LLM output. In contrast, as shown in Ex. 9, when we use TP+TR+TA as the ICL example, the student first follows the TR to create an ER, which subsequently leads to the successful generation of an EA that aligns with the coding style of TA.

---

**Example 7. Code Synthesis without In-context example (without TP, TR, and TA)**

**EP:**
(Leetcode Question ID: 877) Write a python code """class Solution(): def stoneGame(self, piles: List[int]) -> bool:""" to solve the following problem:
Alice and Bob play a game with piles of stones. There are an even number of piles arranged in a row,
...

- - - - - - - - - - - - - - - - - - - - - - - - - - - - - - - - - - - - - - - - - - - - - - - - - -

**EA:**
```
1   from typing import List
2
3   class Solution:
4       def stoneGame(self, piles: List[int]) -> bool:
5           n = len(piles)
6           prefix_sum = [0] * (n + 1)
7           for i in range(n):
8               prefix_sum[i + 1] = prefix_sum[i] + piles[i]
9
10          def dp(i, j):
11              if i > j:
12                  return 0
13              return max(piles[i - 1] + prefix_sum[j] - prefix_sum[i] - dp(i
                        + 1, j),
14                         piles[j] + prefix_sum[i] - prefix_sum[j - 1] - dp(i,
                            j - 1))
15
16          return dp(1, n) \% 2 == 1
```

---

**Example 8. Code Synthesis without TR**

**TP:**
(Leetcode Question ID: 486) Write a python code """class Solution(): def predictTheWinner(self, nums: List[int]) -> bool:""" to solve the following problem:
You are given an integer array 'nums'. Two players are playing a game with this array: player 1 and player 2.
...

- - - - - - - - - - - - - - - - - - - - - - - - - - - - - - - - - - - - - - - - - - - - - - - - - -

**TA:**
```
1   from typing import List
2
3   class Solution:
4       def predictTheWinner(self, nums: List[int]) -> bool:
5           n = len(nums)
6           dp1, dp2 = [0] * n, [0] * n
7           dp1[0], dp2[0] = nums[0], 0
8           for i in range(1, n):
9               dp1[i] = max(dp1[i-1] + nums[i], dp2[i-1])
10              dp2[i] = max(dp1[i-1], dp2[i-1] + nums[i])
11          return dp1[-1] > dp2[-1]
```

- - - - - - - - - - - - - - - - - - - - - - - - - - - - - - - - - - - - - - - - - - - - - - - - - -

**EP:**
(Leetcode Question ID: 877) Write a python code """class Solution(): def stoneGame(self, piles: List[int]) -> bool:""" to solve the following problem:
Alice and Bob play a game with piles of stones. There are an even number of piles arranged in a row,
...

- - - - - - - - - - - - - - - - - - - - - - - - - - - - - - - - - - - - - - - - - - - - - - - - - -

**EA:**

```
1   from typing import List
2
3   class Solution:
4       def stoneGame(self, piles: List[int]) -> bool:
5           n = len(piles)
6           prefix_sum = [0] * (n + 1)
7           for i in range(n):
8               prefix_sum[i + 1] = prefix_sum[i] + piles[i]
9
10          def dp(i, j):
11              if i > j:
12                  return 0
13              return max(prefix_sum[j + 1] - prefix_sum[i] - dp(i + 1, j - 1)
                    ,
14                          prefix_sum[j + 1] - prefix_sum[i] - dp(i, j - 1))
15
16          return dp(0, n - 1) \% 2 == 1
```

## Example 9. **Code Synthesis with TR**

**TP:**

**(Leetcode Question ID: 486)** Write a python code """class Solution(): def predictTheWinner(self, nums: List[int]) -> bool:""" to solve the following problem:

You are given an integer array 'nums'. Two players are playing a game with this array: player 1 and player 2.

...

- - - - - - - - - - - - - - - - - - - - - - - - - - - - - - - - - - - - - - - - - - - - - - - - - - -

**TR and TA:**

The problem can be solved by using a dynamic programming approach. The idea is to maintain two arrays, 'dp1' and 'dp2', where 'dp1[i]' and 'dp2[i]' represent the maximum score that player 1 and player 2 can get respectively after considering the first 'i' elements in the array.

Let's think step by step:

1. Initialize 'dp1' and 'dp2' with zeros. 'dp1[0]' and 'dp2[0]' are set to 'nums[0]' because the first element is the starting point for both players.

2. For each element 'nums[i]' in the array, calculate 'dp1[i]' as the maximum of two options: - 'dp1[i-1] + nums[i]': Player 1 chooses 'nums[i]' and leaves 'nums[i-1]' for player 2. - 'dp2[i-1]': Player 1 chooses 'nums[i-1]' and leaves 'nums[i]' for player 2.

3. Calculate 'dp2[i]' as the maximum of two options: - 'dp1[i-1]': Player 2 chooses 'nums[i-1]' and leaves 'nums[i]' for player 1. - 'dp2[i-1] + nums[i]': Player 2 chooses 'nums[i]' and leaves 'nums[i-1]' for player 1.

4. If 'dp1[-1] > dp2[-1]', return 'True' because player 1 can win. Otherwise, return 'False'.

```
1   from typing import List
2
3   class Solution:
4       def predictTheWinner(self, nums: List[int]) -> bool:
5           n = len(nums)
6           dp1, dp2 = [0] * n, [0] * n
7           dp1[0], dp2[0] = nums[0], 0
8           for i in range(1, n):
9               dp1[i] = max(dp1[i-1] + nums[i], dp2[i-1])
10              dp2[i] = max(dp1[i-1], dp2[i-1] + nums[i])
11          return dp1[-1] > dp2[-1]
```

- - - - - - - - - - - - - - - - - - - - - - - - - - - - - - - - - - - - - - - - - - - - - - - - - - -

**EP:**

**(Leetcode Question ID: 877)** Write a python code """class Solution(): def stoneGame(self, piles: List[int]) -> bool:""" to solve the following problem:

Alice and Bob play a game with piles of stones. There are an even number of piles arranged in a row,

...

- - - - - - - - - - - - - - - - - - - - - - - - - - - - - - - - - - - - - - - - - - - - - - - - - - -

**ER and EA:**

The problem can be solved by using a dynamic programming approach. The idea is to maintain two arrays, 'dp1' and 'dp2', where 'dp1[i]' and 'dp2[i]' represent the maximum number of stones that Alice and Bob can get respectively after considering the first 'i' piles.

Let's think step by step:

1. Initialize 'dp1' and 'dp2' with zeros. 'dp1[0]' and 'dp2[0]' are set to 'piles[0]' because the first pile is the starting point for both players.

2. For each pile 'piles[i]' in the array, calculate 'dp1[i]' as the maximum of two options: - 'dp1[i-1] + piles[i]': Alice chooses 'piles[i]' and leaves 'piles[i-1]' for Bob. - 'dp2[i-1]': Alice chooses 'piles[i-1]' and leaves 'piles[i]' for Bob.

3. Calculate 'dp2[i]' as the maximum of two options: - 'dp1[i-1]': Bob chooses 'piles[i-1]' and leaves 'piles[i]' for Alice. - 'dp2[i-1] + piles[i]': Bob chooses 'piles[i]' and leaves 'piles[i-1]' for Alice.

4. If 'dp1[-1] > dp2[-1]', return 'True' because Alice wins. Otherwise, return 'False'.

```
1   from typing import List
```

```
2
3    class Solution:
4        def stoneGame(self, piles: List[int]) -> bool:
5            n = len(piles)
6            dp1, dp2 = [0] * n, [0] * n
7            dp1[0], dp2[0] = piles[0], 0
8            for i in range(1, n):
9                dp1[i] = max(dp1[i-1] + piles[i], dp2[i-1])
10               dp2[i] = max(dp1[i-1], dp2[i-1] + piles[i])
11           return dp1[-1] > dp2[-1]
```

In the following, we quantitatively assess the following rate of the ICL example and the resulting positive correlation between TA accuracy and the EA accuracy (i.e., LbT score).

- **ICL Example Following Rate (Coding Style).** To quantitatively assess the positive effect of the natural language rationale on ICL following, we define the **Style Follow Rate**, which measures whether EA matches the coding style of TA. Specifically, if the two codes are of the same algorithmic style and raise the same type of exception when an error occurs, we regard them as of the same style. We conduct the judgment manually. For example, if the teacher employs dynamic programming as the problem-solving strategy while the student resorts to a recursive strategy, we classify this as not following. Tab. A15 shows the Style Follow Rate on the *Game Theory* dataset with LLaMA3-8B. **Using both TR and TA as an ICL example yields a 12.5% higher follow rate than using TA alone, illustrating the importance of natural language rationale for enhancing ICL following**.

- **ICL Example Ignore Rate (Coding Style).** Moreover, we observe that when using only the TP+TA as the ICL example, without including TR, LLMs often ignore the ICL example and produce responses that are nearly identical or exactly the same as those generated without any ICL example, as shown in Ex. 8 and Ex. 7. To quantitatively assess this tendency to ignore ICL example in the absence of a natural language rationale, we define the **ICL Ignore Rate**, which measures whether EA's style matches that of the code generated by the student without any ICL example. As shown in Tab. A15, when TR is not used, the ICL Ignore Rate is as high as 43.13%, but using TR reduces this rate significantly to only 1.88%. This further illustrates that **using natural language rationale in the ICL example helps the student more effectively learn from the teacher's code, rather than ignoring it**.

- **Ranking Correlation between TA Accuracy and EPs Accuracy.** We compute Kendall's Tau ranking correlation between the TA's **S-score** and the average **S-score** of EAs, as well as between the TA's **S-score** and the average **V-score** of EAs (i.e., the LbT score). As shown in Tab. A16, we can see that the correlation of "Teach w/ TR+TA" is consistently higher than the corresponding correlation of "Teach w/ TA". When teaching without TR, the correlation between the TA's **S-score** and EAs' **S-score** or **V-score** is zero or even negative.

- **Selected TA Ranking with the Highest EPs Accuracy.** We show the **S-score** ranking of the TA among 8 TAs with the highest average EA score. We observe that: (1) When TPs and EPs are similar and TR is provided in the ICL example, we can select high-accuracy TAs based on the **S-score** or **V-score** of EAs. For example, for questions SG1 and SG4, the TA with the highest EAs' **V-score** achieves the best **S-score** among the 8 TA. (2) When TR is not provided in the ICL example, the **V-score**s of EAs with different TAs as the ICL example are usually not discriminative, i.e., are of exactly the same value. Therefore, when TRs is not provided in the ICL example, using EAs' **V-score** (i.e., LbT score) cannot help select a high-accuracy TA.

Table A15: The Style Follow Rate and ICL Ignore Rate on the *Game Theory* dataset in LeetCode Grandmaster DP study plan with LLaMA3-8B. The *Game Theory* dataset contains 5 problems, and 8 pairs of TR+TA are sampled for each problem. Then, we used the $5 \times 8 = 40$ pairs of TP+TR+TA or TP+TA only to teach students to solve the remaining problems other than TP, resulting in a total of $5 \times 8 \times (5 - 1) = 160$ EAs.

| Method | Style Follow Rate ($\uparrow$) | ICL Ignore Rate ($\downarrow$) |
|---|---|---|
| Teach w/ TR+TA | 81.25% | 1.88% |
| Teach w/ TA | 68.75% | 43.13% |

Table A16: The Kendall's Tau ranking correlation between the TA's **S-score** and the average **V-score** of EAs (i.e., LbT score), and between the TA's **S-score** and the average **S-score** of EAs, and the **S-score** ranking of the TA among 8 TAs ($\in \{1, \cdots, 8\}$, 1 is the best) with the highest average EA score (either LbT score, or the TA's **S-score**). All EPs come from the *Game Theory* dataset. For the "Similar TP and EPs" section, we select TP from the *Game Theory* dataset to ensure high similarity between TPs and EPs. For the "Dissimilar TP and EPs" section, we select TP from the *Tricky Invariant* dataset related to binary search. Details of the teacher IDs can be found in Tab. A8. For each TP, we sampled 8 pairs of TR+TA as the ICL examples.

| TP ID | | Similar TP and EP | | | | Dissimilar TP and EP | |
|---|---|---|---|---|---|---|---|
| | | PW | SG1 | SG3 | SG4 | KMPN | HI2 |
| Teach w/ TR+TA | Kendall's Tau with EAs' **V-Score** | 0.186 | 0.524 | 0.000 | 0.453 | 0.171 | 0.000 |
| | Kendall's Tau with EAs' **S-Score** | -0.074 | 0.725 | 0.000 | -0.074 | 0.356 | -0.371 |
| | TA ranking with max EAs' **V-score** | 1,3 | 1 | 1,3 | 1 | 1,2 | 6 |
| | TA ranking with max EAs' **S-score** | 1 | 2 | 3 | 1 | 1 | 6 |
| Teach w/ TA | Kendall's Tau with EAs' **V-Score** | -0.645 | 0.000 | -0.243 | 0.000 | 0.000 | 0.000 |
| | Kendall's Tau with EAs' **S-Score** | -0.370 | 0.000 | -0.036 | -0.388 | -0.094 | -0.247 |
| | TA ranking with max EAs' **V-score** | 5,7,8 | 1,2,3,4,5,6,7,8 | 2,3,5,7,8 | 1,2,3,4,5,6,7,8 | 1,2,3,4,5,6,7,8 | 1,2,3,4,5,6,7,8 |
| | TA ranking with max EAs' **S-score** | 5 | 8 | 2 | 4 | 6 | 6 |

# B  M2

## B.1  Additional Experimental Setups

We use a learning rate of 5e-7, a batch size of 16 and 1 training epoch. We set $\beta = 0.1$ and add an additional NLL term [55] weighted by 50.

## B.2  Examples

---

**Example 10. A DPO Pair of Correct Answers**

**TP and TP-GT:**
Compute $(-49) \div 7$.
When $y$ is nonzero, we have $(-x) \div y = -(x \div y)$, so $(-49) \div 7 = -(49 \div 7) = \boxed{-7}$.

- - - - - - - - - - - - - - - - - - - - - - - - - - - - - - - - - - - - - - - - - - - - - - - - - - - - -

**Winning TR-TA:**
Let's think step by step.
The division operation is the inverse operation of multiplication, so we need to find the number that 7 multiplied by gives $-49$. This number is $-7$.
Therefore, $(-49) \div 7 = \boxed{-7}$.

- - - - - - - - - - - - - - - - - - - - - - - - - - - - - - - - - - - - - - - - - - - - - - - - - - - - -

**Losing TR-TA:**
Let's think step by step.
To compute division, we can compute the quotient and remainder. We can do this by performing repeated subtraction:

$$49 - 7 = 42$$
$$42 - 7 = 35$$
$$35 - 7 = 28$$
$$28 - 7 = 21$$
$$21 - 7 = 14$$
$$14 - 7 = 7$$
$$7 - 7 = 0$$

At this point, the remainder is 0, so we are done.
The quotient is the number of times we subtracted 7, which is 7. Therefore, $(-49) \div 7 = \boxed{-7}$.

---

**Example 11. A DPO Pair of Wrong Answers**

**TP and TP-GT:**
A positive integer $X$ is 2 more than a multiple of 3. Its units digit is the same as the units digit of a number that is 4 more than a multiple of 5. What is the smallest possible value of $X$?

If a positive integer is 4 more than a multiple of 5, then its units digit must be 4 or 9. We check positive integers ending in 4 or 9 until we find one which is 2 more than a multiple of 3: 4 is 1 more than a multiple of 3, 9 is a multiple of 3, and $\boxed{14}$ is 2 more than a multiple of 3.

- - - - - - - - - - - - - - - - - - - - - - - - - - - - - - - - - - - - - - - - - - - - - - - - - - - - - - -

**Winning TR-TA:**
Let's think step by step.
We know that $X$ is 2 more than a multiple of 3, so $X = 3k + 2$ for some integer $k$.
The units digit of $X$ is the same as the units digit of a number that is 4 more than a multiple of 5. The numbers that are 4 more than a multiple of 5 are 4, 9, 14, 19,..., so the units digit of such a number is either 4 or 9. Since $X$ has units digit 4, we have that $X$ has units digit 4.
Therefore, we can write $X = 3k + 2 = 4$, or $k = 1$. The smallest possible value of $X$ is $\boxed{5}$.

- - - - - - - - - - - - - - - - - - - - - - - - - - - - - - - - - - - - - - - - - - - - - - - - - - - - - - -

**Losing TR-TA:**
Let's think step by step.
The units digit of $X$ is the same as the units digit of a number that is 4 more than a multiple of 5. Since the units digit of a multiple of 5 is either 0 or 5, the units digit of $X$ is either 6 or 1.
Also, $X$ is 2 more than a multiple of 3, so $X$ is either 2 more than a multiple of 3, or 1 more than a multiple of 3.
If the units digit of $X$ is 6, then $X$ must be 8 or 11 or... and so on. However, if $X$ is 2 more than a multiple of 3, then $X$ cannot be 8 or 11 or... because the units digit of 8 or 11 or... is not the same as the units digit of a number that is 4 more than a multiple of 5.
Therefore, the units digit of $X$ must be 1. In this case, $X$ must be 1 or 4 or 7 or... and so on. Since $X$ is 2 more than a multiple of 3, the smallest possible value of $X$ is $\boxed{4}$.

# C M3

## C.1 Workflow

---

**Algorithm A2** The Workflow of **M3**

---

**Require:**

    Teacher and (one or multiple) Student LLM: $\mathbf{T}$, $\mathbf{S}_i$

    Task description $task$, and instances of the task TPs

    Exam Problems: EPs

    Ground-truth of Exam Problems: EPs_GT

    Number of iterations: $T$

1:  # Generate initial exemplars in a zero-shot manner.
2:  exemplars = $\mathbf{T}_{init}(task)$
3:  # Iteratively search for better exemplars.
4:  **for** t = 1 to $T$ **do**
5:     failures = []
6:     # Find student failures when using current exemplars.
7:     **for** EP, EP_GT in zip(EPs, EPs_GT) **do**
8:       **for** $\mathbf{S}_i$ in $\mathbf{S}$ **do**
9:         EA = $\mathbf{S}_i$(exemplars, EP)
10:         **if** EA $\neq$ EP_GT **then**
11:           failures.append((EP, EA, EP_GT))
12:         **end if**
13:       **end for**
14:     **end for**
15:
16:     # Reflect on student failures
17:     reflections = $\mathbf{T}_{reflect}$(exemplars, failures)
18:     # Generate new exemplars based on reflections
19:     **for** i = 1 to **N** **do**
20:       exemplars$_i$ = $\mathbf{T}_{improve}$(exemplars, failures, reflections)
21:       # Evaluate new exemplars
22:       score$_i$ = EVALUATE(EPs_GT, $\mathbf{T}$(exemplars$_i$, EPs))
23:     **end for**
24:     # Update exemplars
25:     best = **ArgMax**(score$_i$)
26:     exemplars = exemplars$_{best}$
27: **end for**
28: # Teacher uses the optimized exemplar to solve Teaching Problem.
29: **return** $\mathbf{T}$(exemplars, TPs)

---

## C.2 Prompt Design

---

**Prompt 6. Teacher Prompt (*init*): Generate Initial Exemplars**

**[User:]**
I'm trying to write $\{k\}$ in-context learning examples for a few-shot classifier. The classifier will answer this question:
"{task}"

Based on the above information, I need a list of $\{k\}$ positive and negative learning examples.
**[Assistant:]**
The list of $\{k\}$ positive and negative learning examples are: {task}

---

**Prompt 7. Teacher Prompt (*reflect*): Reflect on Student Failures**

**[User:]**

---

I'm trying to write {k} in-context learning examples for a few-shot classifier. The classifier will answer this question: "{task}"

But with these examples, the classifier got the following cases wrong: {failure_cases}

Give {num_feedbacks} reasons why these examples could have gotten these cases wrong.

**[Assistant:]**

---

**Prompt 8. Teacher Prompt (*improve*): Improve Exemplars Based on Reflections**

**[User:]**
I'm trying to write {k} in-context learning examples for a few-shot classifier. The classifier will answer this question: "{task}"

But with these examples, the classifier got the following cases wrong: {failure_cases}

Based on these failure cases, the problem with the current in-context examples is that {reflection}.

Based on the above information, I need a new list of {k} improved positive and negative learning examples.

**[Assistant:]**

---

## C.3 Examples

Table A17: Teacher's $F_1$ score of **M3** on Logical Fallacy test set at the end of iteration $T$, where LLaMa3-70B is used as the teacher for all settings. The best results are in **bold**.

| Student(s) | $T = 1$ | $T = 2$ | $T = 3$ | $T = 4$ | $T = 5$ |
|---|---|---|---|---|---|
| LLaMa3-70B | 81.39±0.34 | 81.81±0.32 | 82.10±0.26 | 82.00±0.26 | 82.34±0.22 |
| LLaMa3-8B | **82.44±0.26** | **82.49±0.27** | **82.62±0.24** | 82.72±0.23 | **82.72±0.22** |
| LLaMa3-{70,8}B + Mistral-7B | 82.38±0.28 | 82.48±0.27 | 82.59±0.26 | **82.73±0.24** | 82.29±0.22 |

Ex. 12 provides an instance of how the teacher makes verbalized reflections on why the current in-context learning exemplars are causing students to make mistakes, and how the reflections help guide the teacher to revise the exemplars to address the mistakes.

---

**Example 12. An Example Round of Exemplar Revision**

**Problem:**
Is this text an instance of the logical fallacy of faulty generalization?

Example 1: My father told me that the sky is green. So it must be the truth. **[No]**
Example 2: Which rhetorical fallacy is a broad statement about people on the basis of gender, ethnicity, race, or political, social, professional, or religious group? **[Yes]**
Example 3: But, Dad, I know you had no curfew when you were my age, so how can you give me a curfew? **[No]**
Example 4: Some tall people recently vandalized the park, tall people are irresponsible. **[Yes]**
(some examples omitted)

- - - - - - - - - - - - - - - - - - - - - - - - - - - - - - - - - - - - - - - - - - -

**Student Failures:**
Failure Case 4:
**Input:** We haven't proven aliens didn't create life on earth, so aliens created life on earth.
**Label:** No
**Prediction:** Yes
(some cases omitted)

- - - - - - - - - - - - - - - - - - - - - - - - - - - - - - - - - - - - - - - - - - -

**Reflection:**
The classifier might be having trouble with texts that involve negation or indirect reasoning. In Failure Case 4, the text presents a classic example of an argument from ignorance, but the classifier might have been thrown off by the double negation ("haven't proven...didn't create") or the indirect way of stating the conclusion.

- - - - - - - - - - - - - - - - - - - - - - - - - - - - - - - - - - - - - - - - - - -

**Revised Exemplars:**
Example 3: I've never met a person from France who doesn't love cheese, so the French must have a genetic predisposition to loving cheese. **[Yes]**
Example 7: The fact that some people claim to have seen ghosts doesn't prove that ghosts exist. **[No]**
(some examples omitted)

Table A18: Causes of errors identified by the teacher (LLaMa3-70B) in **M3**, and analysis of whether they also caused teacher mistakes and are mitigated by LbT.

| Student | Cause of student mistakes (identified by teacher) | % teacher mistakes of the same cause | % reduced by LbT |
|---|---|---|---|
| LLaMa3-8B | (1a) Lack of examples within the context of multiple speakers or dialogue; | 45.2% | 6.0% |
| | (1b) Insufficient context for understanding the argument; | 37.1% | 11.6% |
| | (1c) Difficulty in handling nuances of everyday language and humor; | 44.6% | 13.3% |
| Mistral-7B | (2a) Misled by the presence of emotional appeals and excuses in the text; | 60.2% | 0.0% |
| | (2b) Treating a binary or absolute statement as faulty generalization; | 67.2% | 6.5% |
| | (2c) Fail to handle cases involving implicit or indirect relationships between claims and evidence; | 42.5% | 2.3% |
| LLaMa3-70B | (3a) Lack of examples of anecdotal evidence or personal experiences; | 38.2% | 4.5% |
| | (3b) Linguistic structures such as conditional statements; | 83.3% | 0.0% |
| | (3c) Biased towards examples with more complex language or multiple sentences; | 92.4% | 24.1% |

We further analyze the types and proportions of errors made by the students (Tab. A18), and verify that having diverse students help discover a diverse set of errors that the teacher could make, and that our LbT pipeline **M3** could indeed generate exemplars that help correct those errors. More specifically, during one round of exemplar revision for the Logical Fallacy task, we identify the main error types (numbered a,b,c from teacher's reflections of mistakes of different students (numbered 1,2,3).

We observe that the teacher identifies diverse and complementary causes of errors from different students, which help interpret why having diverse students is better. We verify that the causes of students' mistakes (1a, 1b, ...) are indeed also causing teachers' mistakes. Specifically, for each mistake the teacher makes on the test set, we prompt LLaMa-3-70B to judge which cause categories (1a, 1b,...) does this mistake fall into. Note that one mistake can be caused by multiple causes simultaneously. Then, we report the percentage of teacher mistakes of that cause in the third column of Tab. A18. By choosing a student model different from the teacher model, we identify more types of valid causes of teacher mistakes.

Finally, these causes in Tab. A18 indeed help the teacher improve the ICL examples. After the teacher revises the ICL examples by learning from student 1, the teacher's mistakes caused by 1a, 1b & 1c are reduced by 6.0%, 11.6%, and 13.3%. Based on this, we conjecture that LbT methods like **M3** could lead to a more interpretable way to understand the flaws of in-context learning models.

## D  Discussion on Research Rationale

This section discusses our research rationale, and how it leads to specific choices made in our work, including the ultimate **target**, the LbT **idea**, the chosen **tasks**, and the concrete LbT **implementation**. We have already covered the **roadmap** – the next steps for further developing the LbT idea – in § 6, so it is omitted here. Finally, we discuss related machine learning techniques in App. D.5.

### D.1  Target (the "nail" in the high level)

In recent years, we have been impressed by the emergence of powerful AI models with extensive knowledge, strong planning skills, and good intuitions. However, the ability of current LLMs to provide accurate knowledge and reasoning appears to lag behind these other abilities. We consider *accurate knowledge and reasoning* to be among one of the most crucial factors for advancing LLM capabilities.

### D.2 Idea (the "hammer" in the high level)

LLMs demonstrate an impressive ability in grasping solution formats and strategies. However, they often struggle with accurately grasping the underlying logic, differentiating between concepts, and consistently adhering to correct logical reasoning. Actually, these errors mirror some difficulties encountered in human learning. Therefore, we speculate that integrating human educational strategies may help pave the way toward logically accurate AI.

***The LbT methodology has been shown to effectively promote human learning, especially in terms of accurate knowledge building and reasoning.*** Drawing on literature that demystifies the working mechanism of LbT in human learning [5, 18, 22, 33, 62–65] and based on our own teaching and learning experience, we summarize three major ways in which LbT improves human learning:

(a) **Increased self-accountability**: The task of teaching introduces social pressure and incentives, encouraging individuals to raise their standards and work harder.

(b) **Explicit articulation of implicit and vague thoughts**: During the preparation of teaching materials, the teacher needs to use clear and organized *language* to convey its *inner thoughts* to ensure the student's comprehension. This process requires translating implicit concepts into precise terms, describing subtle distinctions, organizing information logically, and so on. This "slow" verbalization and organization not only aids communication, but also helps the teacher identify gaps in its own understanding and discover new connections. As for the teaching material themselves, intuitively, ***teaching materials that make it easier for students to learn have clearer and more accurate logic*** (we refer to this as the "***LbT assumption on teaching material quality***" or the ***LbT-TMQ*** assumption). Especially when the students are unable to solve a set of problems independently, if teaching material 1 enables them to solve the problems more effectively than teaching material 2, it is reasonable to infer that teaching material 1 may have a better logic.

   *We expect **M1** and **M2** to benefit the teacher in a similar way.*

(c) **Iterative feedback from diverse students**: In the teaching process, interaction with students of varying ability levels and knowledge backgrounds offers valuable feedback. The teacher can check if or not students misunderstand the teaching material or struggle with certain problems, and analyze why, in a multi-round discussion. In this process, the teacher might (1) recognize gaps in the teaching material – some conditions and logic may be straightforward to the teacher but require further elaboration and supporting information for students of different backgrounds, to ensure they actually understand and are able to utilize this logic in solving new problems; (2) identify gaps in their own knowledge; (3) discover novel connections when addressing students' misconceptions and erroneous associations.

   *We expect **M3** to benefit the teacher in a similar way.*

We are interested in whether LbT-inspired methods could become a standard prompting or fine-tuning practice to enhance the reasoning abilities of AI models. As a first step, this work investigates whether simple implementations of LbT, focusing on its benefits (b) and (c), can improve the reasoning abilities of contemporary LLMs. For future work, implanting incentives into the LLM learning process to mimic benefit (a) could be a promising pathway, such as setting up a collaborative multi-agent learning framework with proper rewards and communication restrictions.

Additionally, LbT has the potential to improve stronger models by having them teach weaker ones (i.e., weak-to-strong generalization). This might offer exciting opportunities for continuous model evolution, especially as scaling the size of high-quality data faces challenges.

### D.3 Task (the "nail" in the low level)

**Tasks.** In this work, we choose mathematical reasoning, competition-level code synthesis, and verbal logical reasoning, because ***compared to other task domains, these tasks require accurate knowledge and reasoning and cannot be easily solved through vague logic or simple memorization.*** Moreover, all three tasks are popular and have attracted considerable attention from the community.

**Datasets.** For mathematical reasoning, we choose the challenging MATH dataset [27]. First, we experiment with using the `all-mpnet-base-v2` sentence embedding model [60] to calculate the embeddings for all problems. Then, to score the rationale and answer for a given problem, we

select the 2 closest problems based on embedding proximity as the EPs to calculate the LbT score. The results, presented in § 3.3 (right), demonstrate that LbT-based scoring provides substantial relative improvement over SC for problems with close problems. For example, those problems within the smallest 5% cosine distance from their closest problems achieve a 24% relative improvement over SC. However, for problems lacking similar counterparts in the training set, LbT-based scoring may even have a negative effect, which is expected, as the student's score on irrelevant EPs cannot reliably reflect the quality of TRs. MATH() [72] is an extension of the MATH dataset, which offers 3 functional variants for each problem. These variants share a similar solution logic, making the dataset suitable for verifying the LbT-TMQ assumption. Therefore, we choose to conduct the main mathematical reasoning experiments of **M1** on the MATH() dataset.

For code synthesis, we select competition-level problems that require heavy reasoning. Instead of using popular datasets like HumanEval [9] or MBPP [2], which contain mostly basic Python programming problems, we extract more challenging competition-level problems directly from the LeetCode platform. To verify the LbT-TMQ assumption, it is important for the EPs and TPs to be similar. Fortunately, LeetCode offers curated "study plans" (e.g., programming skills, dynamic programming, etc.), each comprising several datasets with problems that exhibit similar underlying structures and solution strategies, making them suitable for our evaluation.

In **M3**, we select the verbal logical reasoning task focused on analyzing and identifying fallacious arguments. In line with our criteria of task selection, determining fallacious arguments also requires accurate reasoning, while the reasoning here takes a verbal natural-language form, complementing the symbolic reasoning in mathematical reasoning and code synthesis. Specifically, we choose Logic [34], a dedicated dataset for logical fallacy detection. In addition, we use Liar [78], a misinformation detection dataset. In misinformation detection, many claims can be factually correct but logically fallacious and it requires LLM's logical reasoning ability on top of factual memorization. For example, the claim "if we allow students to use calculators in math exams, soon they won't be able to perform basic arithmetic without them" is based on the fact that calculators are allowed in some exams, but it is misinformation as it has the logical fallacy of "slippery slope": allowing calculators doesn't necessarily mean students will lose all arithmetic skills, and there is no evidence suggesting such a causal relationship. Logic and Liar are multi-class classification datasets and we cast them as One-vs-Rest binary classification problems following the practice of previous work on prompt optimization [57]. For Logic, we choose the largest class "faulty generalization" as positive and the rest as negative. For Liar, we categorize the class "true" as positive and the rest as negative.

### D.4 Implementation (the "hammer" in the low level)

As a preliminary investigation into the potential of LbT in improving AI, we aim for ***simple implementations*** and ***easily controllable experiments*** to clearly understand the effects of LbT.

- **M1 and M2**: We implement the LbT idea into a single component in well-established pipelines. Specifically, we design an LbT-based scoring component, leveraging the LbT-TMQ assumption "teaching materials that make it easier for students to learn have clearer and more accurate logic". This allows for simple ablation experiments to isolate and understand the benefits derived from this particular LbT-TMQ assumption.

  Baselines: Existing studies have proposed various rationale scoring methods based on GT answer matching [89, 90], answer agreement [31, 79], generation likelihood [79, 84], and self-evaluation [74, 84, 87, 88]. GT answer matching and answer agreement-based methods assume that a more correct or consensus answer indicates a better logic in the corresponding rationale. However, an unclear or wrong rationale can still lead to a correct answer by chance, and a high-quality rationale may result in an incorrect answer due to a simple computational error. In such cases, LbT-based scoring has the potential to better reflect the reasoning quality of the rationale compared to using the correctness and consensus of the final answer.

  Regarding generation likelihood, as [79] found that using the likelihood as the score for weighted self-consistency even degrades the performance, we did not experiment with this method. For self-evaluation, we provide a comparison in Tab. A7, which demonstrates clear improvements with LbT-based scoring over self-evaluation methods.

- **M3**: In addition to implementing the LbT-TMQ assumption as a "static" scoring mechanism for teaching materials, **M3** takes a step forward towards exploring the broader potential of LbT by implementing an "active" and "iterative" feedback process.

### D.5 Relation to other machine learning techniques

**Knowledge distillation.** In traditional task domains, knowledge distillation [23] is a well-known and widely adopted technique. Most knowledge distillation methods aim at improving the student model by transferring knowledge from a fixed stronger teacher. As a special branch in knowledge distillation, mutual learning [92] allows multiple models to learn from each other in an alternating manner. In the realm of LLMs, researchers have explored knowledge distillation [24, 44, 75] and distillation via synthetic data [1, 28, 41] techniques to transfer knowledge from teacher LLMs to student LLMs by letting teacher models *teach* student models through token logits, features, and synthetic data [85]. All of these work follows the "*learning from teachers (LfT)*" instead of our "*learning by teaching (LbT)*" paradigm.

**Meta learning.** Our work also relates to meta-learning [30], a broadly defined term encompassing various methods within the "better learning by *learning to learn*" paradigm. These approaches typically aim to optimize different aspects of the learning process, such as initial weights [21], training hyperparameters (e.g., learning rate, regularization coefficient) [20, 70], neural architectures [19], data augmentations [15], and so on. Many approaches formulate meta-learning as a bilevel optimization problem, where the outer optimization targets the configuration of interest, and the inner optimization is the learning process itself. In the era of increasingly intelligent LLMs, our work introduces a novel, human cognition-inspired paradigm – "better learning by *learning to teach*" (LbT) – to continuously evolve the LLMs, which uses the "task" of teaching (weaker) models to challenge the LLM. Initially, we considered a straightforward implementation involving explicit bilevel optimization, where the inner optimization is the student's learning process, and the outer optimization can be conducted with reinforcement learning techniques. However, by leveraging the in-context learning abilities of LLMs, we were able to explore the LbT paradigm in a simpler and controllable manner. For example, we optimize the ICL examples for the teacher itself by observing students' results in **M3**.

**Machine teaching.** Another related area is machine teaching [7, 97, 98], which is defined as an inverse problem of machine learning: finding the optimal teaching examples if the teacher already knows the parameters, and aims to make the *student* learn the target concept with a minimal set of labeled examples. In contrast, our work focuses on enhancing the performance of the *teacher*, as measured by the teacher's accuracy on a test set. In addition, while machine teaching literature typically focuses on small-scale tasks using analytical models, our focus is on the reasoning abilities of state-of-the-art LLMs.

**Other methods related to the LbT idea.** It is worth noting that an independent work [32] designs a regularization loss to improve model generalization. This regularization loss can help filter out extraneous details that are difficult for auxiliary students to imitate. This approach is based on the assumption that "*generalizable correlations should be easy to imitate*", which shares a conceptual similarity with our LbT-TMQ assumption. Another recent work, Math-Shepherd [77], proposes an automatic scoring method for partial rationales within the generating-scoring-finetuning pipeline. It evaluates each partial rationale by measuring how often another "completer" model arrives at the correct answer by continuing from the partial rationale. Although this paper does not frame the method from an LbT perspective, the method can be interpreted as LbT-based scoring, based on the assumption that "*partial rationales which make it easier for students to continue and reach the correct answer have clearer and more accurate logic*". In Math-Shepherd, the students (i.e., the completer) are examined by extending the partial teaching rationale for the same problem, whereas in our **M2**, students are examined on similar problems, using the full rationale from the teaching problem as an exemplar. We hope that framing this method together with our methods within the LbT framework could help enlighten a broader roadmap for drawing insights from human education.

## E More Extension Possibilities

Based on the framework presented in § 1 and 2, we can explore other possibilities to incorporate LbT insights into the inference and training of LLMs. For example, to improve answer quality for a TP, we can incorporate the idea of having the teacher reflect on multiple students' feedback into the search-based output generation pipeline, as follows: for each student, the teacher iteratively reflects on the student's feedback and updates its TR. The final answer is then derived by aggregating the updated TR for each student. Intuitively, different students might require distinct teaching materials, so generating TR for multiple students can encourage TR diversity. Increasing diversity has been

shown to benefit the accuracy of agreement-based aggregation methods [79]. Compared to increasing the sampling temperature of TR, this LbT-based sampling is guided towards high-quality TRs and might yield better results. Besides, LbT can potentially be extended to open-ended problems, such as dialogue, writing, and open-ended math problems [86]. A natural extension could involve using a teacher LLM to evaluate a student's answer [10] and provide the LbT score.

## F   Potential Risks of Bias Perpetuation

In open-domain problems where no GT judgment exists and LLM-based judgment is needed, it is possible that teaching materials deemed "well accepted" by students are not necessarily more accurate or closer to the truth. Instead, they may align with the existing biases of teachers or students, posing a risk of the teacher perpetuating their own biases [68] or indirectly learning the students' biases.

Furthermore, while this work primarily focuses on utilizing the LbT idea to improve reasoning in mathematical and code domains, considering the potential bias perpetuation effect becomes even more critical in domains where societal bias and fairness are of paramount concern [76].

