# OpenReview forum: "Can LLMs Learn by Teaching for Better Reasoning? A Preliminary Study"
_NeurIPS.cc/2024/Conference — NeurIPS 2024 poster_

### Official Review · Reviewer_E9eU · 2024-07-01

**Soundness:** 3
**Presentation:** 3
**Contribution:** 3
**Rating:** 5
**Confidence:** 5

**Summary:**

The core question raised in the paper is whether LLMs can also learn through teaching (LbT). The authors demonstrate that the idea of LbT can be easily integrated into existing LLM training/prompting processes and propose three methods, each mimicking three levels of human LbT: observing feedback, learning from feedback, and iterative learning. The authors found that: (1) LbT can induce a progression from weak to strong generalization, where a strong model can improve itself by teaching other weak models; (2) Diversity among students is important, as teaching multiple students may be better than teaching just one student or the teacher itself.

Corresponding solutions:

1）M1 aims to improve the answer quality of LLMs by directly leveraging students' feedback (L1). 2） M2 aims to enhance the intrinsic abilities of LLMs by learning from students' feedback (L2). 3） M3 aims to improve the answer quality of LLMs by iteratively learning from students' feedback (L3).

Strengths: The paper provides a clear and detailed description of the problem and the proposed solutions, supported by ample experimentation.

Weakness:

The idea presented in the article is a well-known idea, but it lacks sufficient assumptions and counterexample analysis, making it too intuitive. See:
[1]Black-box Generalization of Machine Teaching, Xiaofeng Cao, Yaming Guo, Ivor W. Tsang, James T. Kwok, https://arxiv.org/abs/2206.15205v2.

Employing teaching feedback to guide a white-box or black-box learner has been mentioned earlier.

In machine learning, such an idea is the most basic one, especially in machine teaching. See:
[2] Liu, W., Dai, B., Humayun, A., Tay, C., Yu, C., Smith, L. B., ... & Song, L. (2017, July). Iterative machine teaching. In International Conference on Machine Learning (pp. 2149-2158). PMLR.

[3] Zhang, C., Cao, X., Liu, W., Tsang, I., & Kwok, J. (2023, July). Nonparametric iterative machine teaching. In International Conference on Machine Learning (pp. 40851-40870). PMLR.

There are also works for teaching multiple learners: [4] Yeo, T., Kamalaruban, P., Singla, A., Merchant, A., Asselborn, T., Faucon, L., ... & Cevher, V. (2019, July). Iterative classroom teaching. In Proceedings of the AAAI Conference on Artificial Intelligence (Vol. 33, No. 01, pp. 5684-5692).

[5] Zhang, C., Cao, X., Liu, W., Tsang, I., & Kwok, J. (2024). Nonparametric teaching for multiple learners. Advances in Neural Information Processing Systems, 36.

The authors lack a survey of the machine learning community regarding teaching.

The author exaggerates their contribution or should strengthen the underlying assumptions of their contribution. Teaching weak learners is related to their variance.

Teaching weak students is challenging unless there is a proper configuration. Otherwise, the improvement in students' learning and reception abilities is limited, as already verified in knowledge distillation (KD).

5）We have only seen separate presentations of the M1-M3 approaches. It is unclear how M1-M3 collectively contribute to the overall logic and objectives of the paper.

6）M3 mentioned in the paper is not sufficient to be considered a core contribution.

7）Would these methods still be effective if there is a significant difference between the student model and the teacher model?

Overall, this draft studies the teaching in the current LLM settings. While the assumptions and conclusions have issues based on current presentation. I suggest the authors tune down their statements.

**Strengths:**

Strengths:
The paper provides a clear and detailed description of the problem and the proposed solutions, supported by ample experimentation.

**Weaknesses:**

Weakness:

The idea presented in the article is a well-known idea, but it lacks sufficient assumptions and counterexample analysis, making it too intuitive. See:
[1]Black-box Generalization of Machine Teaching, Xiaofeng Cao, Yaming Guo, Ivor W. Tsang, James T. Kwok, https://arxiv.org/abs/2206.15205v2.

Employing teaching feedback to guide a white-box or black-box learner has been mentioned earlier.

In machine learning, such an idea is the most basic one, especially in machine teaching. See:
[2] Liu, W., Dai, B., Humayun, A., Tay, C., Yu, C., Smith, L. B., ... & Song, L. (2017, July). Iterative machine teaching. In International Conference on Machine Learning (pp. 2149-2158). PMLR.

[3] Zhang, C., Cao, X., Liu, W., Tsang, I., & Kwok, J. (2023, July). Nonparametric iterative machine teaching. In International Conference on Machine Learning (pp. 40851-40870). PMLR.

There are also works for teaching multiple learners: [4] Yeo, T., Kamalaruban, P., Singla, A., Merchant, A., Asselborn, T., Faucon, L., ... & Cevher, V. (2019, July). Iterative classroom teaching. In Proceedings of the AAAI Conference on Artificial Intelligence (Vol. 33, No. 01, pp. 5684-5692).

[5] Zhang, C., Cao, X., Liu, W., Tsang, I., & Kwok, J. (2024). Nonparametric teaching for multiple learners. Advances in Neural Information Processing Systems, 36.

The authors lack a survey of the machine learning community regarding teaching.

The author exaggerates their contribution or should strengthen the underlying assumptions of their contribution. Teaching weak learners is related to their variance.

Teaching weak students is challenging unless there is a proper configuration. Otherwise, the improvement in students' learning and reception abilities is limited, as already verified in knowledge distillation (KD).

5）We have only seen separate presentations of the M1-M3 approaches. It is unclear how M1-M3 collectively contribute to the overall logic and objectives of the paper.

6）M3 mentioned in the paper is not sufficient to be considered a core contribution.

7）Would these methods still be effective if there is a significant difference between the student model and the teacher model?

Overall, this draft studies the teaching in the current LLM settings. While the assumptions and conclusions have issues based on current presentation. I suggest the authors tone down their statements.

**Questions:**

a) Would these methods still be effective if there is a significant difference between the student model and the teacher model?

b) Teaching weak students is challenging unless there is a proper configuration. Otherwise, the improvement in students' learning and reception abilities is limited, as already verified in knowledge distillation (KD).

**Limitations:**

Incomplete analysis from the teaching and irregular contribution statements.

---

> ### Author Rebuttal · Authors · 2024-08-07
>
> **Q1: The idea presented in the article is a well-known idea.**
>
> Thank you for suggesting relevant references from the machine teaching literature. We will cite and discuss them in our revised version. While these papers have similarities in terms of how the teacher should organize teaching materials, our work differs from prior studies in two key aspects:
> 1. As stated in our introduction, "teaching" has been extensively studied in machine learning, with knowledge distillation as a prominent example. While both machine teaching and knowledge distillation aim to **improve the student**, our focus is on whether using "teaching" to get students' feedback can help **improve the teacher**—especially in the context of weak-to-strong generalization (i.e., continuously evolve *stronger* teacher by teaching *weaker* students). This distinction sets our paper apart from existing work.
> 2. Instead of solving small-scaled tasks using simple analytical models, such as linear or kernel models, as is common in much of the machine teaching literature, our work focuses on advancing the **reasoning capabilities of contemporary LLMs** with tens of billions of parameters. As LLMs get stronger and stronger and even show human-like behaviors nowadays, our work aims to evaluate whether the LbT methodology that has proven effective in providing intrinsic supervision for accurate knowledge building and reasoning in human learning can similarly benefit LLMs.
>
> To summarize, as far as we are aware, this is the first attempt at migrating the LbT idea from the learning sciences [3, 4, 5, 6, 7, 8, 9, 10] to enhance LLMs. We will discuss these key differences, including the motivation, the model scale, and the targeting tasks, between our work and provided references from the machine teaching community in the revision.
>
> **Q2: It lacks sufficient assumptions and counterexample analysis, making it too intuitive.**
>
> Thanks for this comment. We'd like to share our perspectives on this issue. We acknowledge that since our study focuses on a complex and powerful LLM solving complex reasoning problems, it is challenging to derive comprehensive theoretical frameworks with precise assumptions and counterexamples. As LLMs are becoming stronger and even show human-like behaviors nowadays, we believe that advancing algorithms at both ends of the spectrum can help push the field forward:
> - Developing methods and models grounded with theoretical guarantees to solve simple or synthetic tasks.
> - Building intuitive methods for use with state-of-the-art models to address real-world complex tasks. These methods, inspired by human problem-solving and learning processes, are often empirically validated through experimental analyses rather than being grounded in theoretical frameworks.
>
> There are many other studies that also draw inspiration from human problem-solving techniques to enhance their capabilities, which have created enormous impact on the LLM community. For instance, techniques such as Chain-of-Thought [24] is motivated by the way humans reason through problems step-by-step. Self-Refine [25] is inspired by the human practice of refining written text. Similarly, Relexion [26] emulates how humans iteratively learn and adapt when tackling complex tasks. These works, together with ours, lie in the second category of the spectrum.
>
> **Q3: Teaching weak learners is related to their variance.**
>
> We are not very sure what "variance" refers to in this context. Could you please provide more details on this comment? We are eager to understand your perspective better and address any concerns you might have.
>
> **Q4: Teaching weak students is challenging unless there is a proper configuration. Otherwise, the improvement in students' learning and reception abilities is limited, as already verified in knowledge distillation (KD).**
>
> **Q7: Would these methods still be effective if there is a significant difference between the student model and the teacher model?**
>
> Thank you for the great question. About why teachers can benefit from teaching weak students, our intuition is as follows. To teach a weak student, the teacher needs to organize detailed and high-quality teaching materials that can be digested even by a weak student. This is a high bar for the teachers and the teachers should be able to learn a lot from material preparation and the feedback from the students. However, the students should not be too weak. Otherwise, they may lack the ability to learn anything from the teacher (e.g. through in-context learning), and therefore cannot provide useful feedback.
>
> Indeed, the effectiveness of LbT varies as the teacher-student configuration changes. As demonstrated by our experiments, certain configurations are less effective than others. For example, as shown in Table 2, when ChatGPT-3.5 is the teacher, it is more effective to use LLaMA3-8B as the student than Mistral-7B.
>
> Despite this, our method proves effective even with a significant disparity between the teacher and the student. For example, in M1 and M3, we have shown that LLaMA3-70B (70.16% on 181 MATH) can benefit from teaching LLaMA3-8B (45.85% on 181 MATH). In M1, we demonstrated that GPT-3.5 (59.11% on 181 MATH) can benefit from teaching Mistral-7B (19.88% on 181 MATH). These examples involve models from different families with a large capacity gap.
>
> To further demonstrate the robustness of our method across various configurations, we conducted an additional experiment where LLaMA3-70B teaches Mistral-7B. Please refer to the additional results in the provided PDF for more details.
>
> Regarding the comment "the improvement in students' learning and reception abilities is limited", we are not sure whether we understand this comment correctly, but we aim at improving the performance of the **teacher** rather than the **student**. Please do not hesitate to correct our understanding if necessary.

---

> ### Author Response · Authors · 2024-08-07
> **Additional rebuttal 1**
>
> **Q5: We have only seen separate presentations of the M1-M3 approaches. It is unclear how M1-M3 collectively contribute to the overall logic and objectives of the paper.**
>
> The relationship between M1-M3: M2 is built on top of M1, where we derive the LbT scores from M1 and use them to fine-tune the teacher in M2. The combination of M3 with M1 is discussed in lines 268-277 as a near-term extension.
>
> How M1-M3 collectively contribute to the overall logic and objective: Our aim is to study whether the general idea of LbT can help improve the crucial reasoning ability of LLMs. We summarize LbT in human learning into three levels (which is backed up by observations from the learning sciences [3-12]) and then demonstrate its effectiveness through three case studies, each corresponding to one of the three LbT levels. The exact objectives of each concrete method design are also summarized in Table 1.
>
> **Q6: M3 mentioned in the paper is not sufficient to be considered a core contribution.**
>
> We acknowledge that the idea of organizing teaching materials shares similarities with existing literature, particularly within the machine teaching community, but as discussed in the previous response, our focus on improving the ability of the **teacher** using LLMs is distinct from previous studies.
>
> Furthermore, our aim is not to design brand-new pipelines, but to illustrate how the LbT idea can be integrated into existing LLM training and prompting pipelines. This integration offers exciting opportunities for models to evolve by teaching other (potentially weaker) models. The connection between our methods and existing pipelines is discussed in Section 2.

---

> ### Comment · Reviewer_E9eU · 2024-08-09
> **Please tone donw the statements**
>
> ----"The findings are rather encouraging. For example, similar to LbT in human, we see that: (1) LbT can induce weak-to-strong generalization: strong models can improve themselves by teaching other weak models; (2) Diversity in students is important: teaching multiple students could be better than teaching one student or the teacher itself. "
>
>
> ----"However, the students should not be too weak. Otherwise, they may lack the ability to learn anything from the teacher (e.g. through in-context learning), and therefore cannot provide useful feedback."
>
>
> Those statements have issues, but the authors don't tone down them. For teaching, "teaching a black-box learner" is feasible, and it is a special track in the machine learning community. "Teaching multiple students could be better than teaching one student" is also wrong. The students' variance in learning ability affects the iterative teaching performance. There are also topics where weak teachers teaching heterogeneous students, i.e., classroom teaching, is applicable. I suggest the authors read more works from Jerryzhu and Sanjoy Dasgupta.
>
> Considering the authors present too many inaccurate statements in the draft and don't realize the proposed issues, I have to tune down my score.
>
> Although the author was working on an engineering project, they could not deviate from the basic theoretical statements to avoid exaggeration and false demonstration of certain issues.

---

> > ### Author Response · Authors · 2024-08-10
> > **We appreciate the reviewer for the prompt reply. Here are the follow-up discussions (part I).**
> >
> > We appreciate the reviewer for the prompt reply. It seems like the remaining concern lies in the tone of the statements. **We are more than willing to revise statements that may have tone issues.** Please allow us to provide more discussions regarding this concern.
> >
> > ---
> > > **General Comment: Those statements have issues, but the authors don't tone down them.
> > Considering the authors present too many inaccurate statements in the draft and don't realize the proposed issues, I have to tune down my score.
> > Although the author was working on an engineering project, they could not deviate from the basic theoretical statements to avoid exaggeration and false demonstration of certain issues.**
> >
> > We definitely agree that any research work should avoid exaggeration. In the paper as well as the rebuttal, we carefully stated that our scope is to conduct preliminary exploration of **LbT** in **LLMs for reasoning tasks**, and our conclusions are confined to the empirical results of our experiments. We did not mean to exaggerate that these observations are held generally in other domains. For example, in the title and abstract, we stated the scope clearly "Can LLMs also learn by teaching (LbT)? ... In this paper, we provide a preliminary exploration of this ambitious agenda." before presenting the findings; when describing the results, we use "could/can" when possible. We went through the paper again to double-check our statements. We will revise the tone of two short summary sentences according to the reviewer's suggestion (see the reply to Follow-up Comment 1), and we did not spot other issues beyond that.
> >
> > Regarding the deviation from the theoretical statements and citations the reviewer brought up, we want to emphasize that their context and settings are quite different from ours. Thus, their results do not necessarily generalize to our problem setting, and their different results are not sufficient to claim that our results are wrong (see the reply to Comment 4).
> >
> > BTW, we are happy to provide the code for reproducing all empirical results described by our statements. According to this year's Neurips policy, authors are not allowed to post links without reviewers' request. If needed, we will provide an anonymous code link if the reviewer could submit a code request comment.
> >
> > > **Follow-up Comment 1 -- A quote of two statements in the abstract.**
> >
> > Regarding the statement "Diversity in students is important: teaching multiple students could be better than teaching one student or the teacher itself". We have chosen the word "could" in the statement "... could be better ..." to describe the empirical results. We will further tone down the statement from "Diversity in students is important" to "Diversity in students might help" according to the suggestion.
> >
> > Regarding the statement "LbT can induce weak-to-strong generalization: strong models can improve themselves by teaching other weak models". We think the statement is correct as it already uses "can" and describes the empirical results under our settings. We can further tone the summary down from "LbT can induce weak-to-strong generalization" to "LbT might help with weak-to-strong generalization".
> >
> > We have checked our paper again. Except for these two sentences (which have corresponding explanations with proper tones), we think all sentences are in the proper tones. If there are other statements that could lead to potential misunderstandings, please let us know and we will revise them.
> >
> > > **Follow-up Comment 2: For teaching, teaching a black-box learner is feasible, and it is a special track in the machine learning community.**
> >
> > We guess that this comment is about this statement in our original rebuttal: "However, the students should not be too weak. Otherwise, they may lack the ability to learn anything from the teacher (e.g. through in-context learning), and therefore cannot provide useful feedback".
> >
> > Does the reviewer mean that "teaching a black-box student is feasible, even if it is weak"? If so (if not, please let us know), we'd like to emphasize that this statement in the rebuttal describes the findings under our settings, i.e., LLM for reasoning task, with in-context learning as the "teaching method". The detailed logic is as follows: We employ in-context learning to let the student learn from the teacher's output (using the TP and teacher's TR as the in-context exemplar). If the student is too weak to follow the in-context learning exemplars to solve Exam Problems, then its score on the Exam Problems cannot reflect the quality of the teaching material, i.e., failing to provide useful feedback to further guide the teacher's generation or training.
> >
> > If we have any misunderstandings about this comment, please let us know!

---

> ### Author Response · Authors · 2024-08-10
> **We appreciate the reviewer for the prompt reply. Here are the follow-up discussions (part II).**
>
> > Follow-up Comment 3&4: "Teaching multiple students could be better than teaching one student" is also wrong. -- with two supporting comments.
>
> Again, this statement was accurately describing our empirical findings in M1 and M3. We discuss the two supporting comments one as follows:
>
> > Comment 3: The students' variance in learning ability affects the iterative teaching performance.
>
> Based on this comment, **we think the misunderstanding might come from the definition of the term "teaching performance"**. For clarification, in machine teaching literature, "teaching performance" refers to the **student's accuracy**, whereas "teacher performance" in our work refers to the **teacher's accuracy**.
>
> In our experiments, under the settings of teaching multiple students and teaching one student, we both report the **teacher's final generation accuracy** with M1/M3. Based on the previous clarification, we think making this claim "Teaching multiple students could be better than teaching one student" is direct and valid from this empirical comparison.
>
> It might be possible that we still do not fully understand what your comment means. We have also asked for clarification in our original rebuttal: "We are not very sure what variance refers to in this context. Could you please provide more details on this comment? We are eager to understand your perspective better and address any concerns you might have". Could the reviewer explain it in more detail? We will be happy to discuss it further.
>
> > Comment 4: There are also topics where weak teachers teaching heterogeneous students, i.e., classroom teaching, is applicable. I suggest the authors read more works from Jerryzhu and Sanjoy Dasgupta.
>
> Thanks again for recommending the work from these researchers. We have already gone through all the literature recommended in the original review. We have discussed the connections between the prior works and our paper in the response to **Q1**. While we find these works from the machine teaching community to be highly inspiring, there are key differences between their focus and ours:
> - Machine teaching is defined as *"an inverse problem of machine learning, that is, finding the optimal teaching examples if the teacher already knows the learning parameters."* [27, 28] and aims to improve the performance of the **student** with a minimal set of labeled examples. In contrast, our work focuses on enhancing the performance of the **teacher**, as measured by the **teacher's accuracy** on a test set.
> - Machine teaching typically focuses on **small-scaled tasks using analytical models**, whereas we focus on the **reasoning capabilities of contemporary LLMs**.
>
> Therefore, we respectfully argue that it is not appropriate to claim our statements are wrong based on literature and results from a significantly different context.
>
> That being said, we agree that acknowledging these works in our paper will provide valuable context. Thank the reviewer for pointing them out. As we promised in our rebuttal, "We will discuss these key differences, including the motivation, the model scale, and the targeting tasks, between our work and provided references from the machine teaching community in the revision."
>
> ---
> Thank the reviewer again. As criticisms and discussions are very helpful for enhancing the quality of our paper, we are happy to discuss further!
>
> [27] An Overview of Machine Teaching. arXiv, 2018.
>
> [28] Black-box Generalization of Machine Teaching. arXiv, 2022.

---

> > ### Comment · Reviewer_E9eU · 2024-08-13
> > **Lack of rigor and distinctive contributions**
> >
> > It is beneficial that the authors are able to approach the concept of teaching from a theoretical perspective, as this can provide a more rigorous logic for the paper. Teaching multiple students is challenging, as the variance among students can affect overall performance, even with effective communication skills.
> >
> > LLM is a new topic in the community, so introducing teaching as a concept is a reasonable approach. However, from an optimization standpoint, the contributions related to using teaching feedback to improve performance are not novel. As mentioned in teaching theory, this is a well-known paradigm. Similarly, in knowledge distillation (application of teaching), this concept is also not new. Therefore, it is difficult to identify the unique contributions of the draft.
> >
> > I hope the authors can enhance the quality of their ideas by seeking more stringent assumptions. At the very least, the paper should address issues relevant to the machine learning community.
> >
> > Academic rigor should be prioritized over incremental improvements in results.
> >
> > I will maintain my initial score of Borderline Accept. Thank you for the positive discussion.

---

> > > ### Author Response · Authors · 2024-08-13
> > >
> > > Thank you for raising the score. We will ensure that our revision includes a discussion on the differences between our work from the machine teaching literature.

---

### Official Review · Reviewer_mNjM · 2024-07-13

**Soundness:** 3
**Presentation:** 4
**Contribution:** 3
**Rating:** 7
**Confidence:** 4

**Summary:**

In this paper the authors investigate whether the principles of 'Learning by Teaching' (LbT) in humans can be applied and used in LLMs. To investigate this they propose 3 techniques and map them different to LbT levels.

The first technique M1 aims at improving answer quality by developing a scoring function to rank answers generated by LLMs. For the purpose of this method a strong LLM is made to generate Teaching Rationale (TR) and Teaching Answer (TA) pairs for a given teaching problem (TP). The TR-TA pairs are used as ICL exemplars within student models presented with an Exam Problem (EP) which is of similar type as the TP.  The student models then produce Exam Rationale (ER) and Exam Answers (EA) and receive an accuracy score or LbT score based on the correctness of EA. The TR-TA pairs are then selected based on the highest score and the highest sum of scores. The authors demonstrate that this scoring strategy outperforms self-consistency scoring/greedy scoring in a variety of teacher-student model settings and a variety of tasks (e.g. Math, Coding)

The second technique M2 aims to improve the ability of LLMs by leveraging the student feedback. For this DPO is used to finetune the teacher model on the TR and the corresponding LbT scores obtained using M1 along with correctness scores. The authors show that this technique leads to better performance than by simply doing DPO with the correctness scores.

The third technique M3 is based on iteratively improving the exemplars generated for the students by reflecting on the mistakes made by the students. The authors demonstrate that this technique helps improve the performance and also benefits when multiple student LLMs are used to provide feedback.

In summary the authors conclude by suggesting that strong teacher models can improve even when teaching weak students(weak-to-strong generalization) and teaching other students/multiple students works better than teaching itself.

**Strengths:**

The paper is well written and provides several interesting and novel results which could be of significant interest to the research community (particularly alignment). The experimental analysis and discussion of results is sound and well supported by evidence.  Along with the introduction of LbT the paper provides concrete methods (M1, M2, M3) to instantiate different aspects of LbT in LLMs. The paper also presents some encouraging findings such as 1) LbT can be used to improve answer quality and model capability, 2) LbT exhibits weak-to-strong generalization and 3) diversity of student models helps.

**Weaknesses:**

The LbT score seems to be reliant on having the final answer being verifiable. It would be interesting to see how this can be translated to cases where the answer produced by the student models is in free-form text. In this regard the current use-cases demonstrated have been in improving performance on tasks such as math or coding. More extensive evaluation on diverse tasks may be needed to assess the generalizability of LbT. A more detailed error analysis on what types of errors the student models make and which of these errors provide the most helpful feedback to the teacher model could be interesting.

**Questions:**

For the claim that 'Improvements do not saturate as number of TR-TA pairs increase' has there been any analysis done to identify the upper bound or the optimal number of TR-TA pairs needed (cost vs performance)?

General Comments:
There is a typo on line 53.

**Limitations:**

The authors acknowledge that the technique requires the strategies to solve the EPs be similar to the ones used for the TPs and that currently the EPs were selected based on human provided information.
As the authors acknowledge the proposed technique also leads to additional inference cost.
Furthermore it remains to be seen how this technique can be extended to tasks wherein the answers generated by student models are not easily verifiable.

---

> ### Author Rebuttal · Authors · 2024-08-07
>
> **Q1: The LbT score seems to be reliant on having the final answer being verifiable. ... More extensive evaluation on diverse tasks may be needed to assess the generalizability of LbT.**
>
> Thanks for this valuable question. LbT can indeed be extended to open-ended problems, such as dialogue, writing, and open-ended math problems. A natural extension could involve using a teacher LLM to evaluate a student's answer, which would then serve as the LbT score. This approach mirrors how a teacher assesses a student in human learning and is promising due to the strong evaluation capabilities of LLMs [19, 20]. A similar idea has been employed to generalize Self-Consistency (which only applies to verifiable problems) [21] to open-form problems [22].
>
> Besides, as demonstrated in literature from the learning sciences, teachers may directly benefit from providing assessments [8] or reviews [23], and students can provide valuable free-form feedback beyond taking tests or exams, such as peer recommendations [11] and satisfaction questionnaires [16, 17]. We believe that an open-ended evaluation process could potentially offer additional feedback to the teacher LLM, which can be utilized to further enhance its performance. We plan to explore these extensions in future studies.
>
> **Q2: A more detailed error analysis on what types of errors the student models make and which of these errors provide the most helpful feedback to the teacher model could be interesting.**
>
> Thanks for this valuable suggestion. For math reasoning (M2), we have included some analyses in Section 4.3 and presented the examples in Appendix B.2. For competition-level code synthesis (M1), we have included analyses in Section 3.3.2 and presented the examples in Appendix A.2.2.
>
> For math reasoning and code synthesis, students can make both logical and non-logical errors. The non-logical errors, such as computation errors (Math), missing imports (Code), miswritten variable names (Code), and incorrect usage of library functions (Code), are mainly related to the knowledge required by specific EQs and the robustness of the student model, rather than reflecting the quality of the TR-TAs. Thus, they do not provide helpful feedback to the teacher. To address this, for code synthesis, we apply self-debugging to correct non-logical bugs, making the students' exam V-score a more accurate indicator of TR quality and thus more helpful to the teacher.
>
> In response to your suggestion, we will add more detailed analyses and examples to the appendix. For example, logical errors in code synthesis can be further classified into three types: (1) Code with generally correct logic but incorrect handling of boundary conditions, which fails in some hard cases; (2) Code with incorrect logic, such as a wrong DP recursion formula, which typically fails most cases; and (3) Code with correct logic but complexity issues, such as using recursion instead of DP or cached-recursion, leading to time or memory errors on large cases. Our experiments show that when students follow TR-TA in making these three types of logical errors, it provides valuable feedback for the teacher.
>
> For textual reasoning tasks (M3), according to your suggestion, we add more error analysis in the attached PDF. More specifically, we further analyzed M3's behavior in identifying false generalizations [18] with Llama-3-70B as the teacher and Llama-3-8B as the student. Some causes of errors identified by the teacher from student mistakes are: (a) "Lack of examples within the context of multiple speakers or dialogue"; (b)"Insufficient context for understanding the argument"; (c) "Difficulty in handling nuances of everyday language and humor". First, we found that the errors that the teacher identifies from student mistakes are **also applicable to teacher's mistakes**, with 45.2%, 37.1%, and 44.6% of teacher mistakes caused by these three reasons. Second, after the teacher improves the ICL examples by learning from the students, the teacher's mistakes caused by these reasons are reduced by 6.0%, 11.6%. and 13.3%. Finally, mistakes of different students lead to complementary causes of errors that are also very relevant to teacher mistakes.
>
> **Q3: For the claim that 'Improvements do not saturate as number of TR-TA pairs increase' has there been any analysis done to identify the upper bound or the optimal number of TR-TA pairs needed (cost vs performance)?**
>
> Thanks for raising this valuable question. In the rebuttal, we have extended our M1 Math experiments to include 256 TR-TA pairs. Please refer to the updated Figure 4 (Left) in the provided PDF. While the curve does become flatter with the increase in the number of TR-TA pairs, we still observe a non-negligible slope even with 256 TR-TA pairs.
>
> For precision, we will revise this claim to: "The relative improvement over SC increases as the number of TR-TA pairs increases within the range of TR-TA pairs in our experiments".

---

> > ### Comment · Reviewer_mNjM · 2024-08-11
> > **Acknowledgment of rebuttal**
> >
> > I thank the authors for the clarifications. I am keeping my score

---

> > > ### Author Response · Authors · 2024-08-11
> > >
> > > Thank you very much for your great questions and support!

---

### Official Review · Reviewer_pWvs · 2024-07-14

**Soundness:** 3
**Presentation:** 4
**Contribution:** 3
**Rating:** 7
**Confidence:** 3

**Summary:**

This paper the use of learning by teaching methods in the context of LLMs.

**Strengths:**

The paper is well written and methodologically sound.

**Weaknesses:**

The concise results should be briefly and systematically stated in the final Conclusion chapter, which is missing.

**Questions:**

N/A

---

> ### Author Rebuttal · Authors · 2024-08-07
>
> **Q1: The concise results should be briefly and systematically stated in the final Conclusion chapter, which is missing.**
>
> Thank you for the suggestion. We will add a conclusion section and summarize the concise numbers together with the general conclusion there.

---

### Official Review · Reviewer_UmKq · 2024-07-17

**Soundness:** 4
**Presentation:** 4
**Contribution:** 4
**Rating:** 7
**Confidence:** 4

**Summary:**

This paper "Can LLMs Learn by Teaching? A Preliminary Study" presents a novel approach towards LLM learning by teaching with three methods: observing student feedback, learning from student feedback, and learning iteratively. The contribute two key findings: teaching student models are an effective way to improve model performance (with fine-tuning), and student models must be diverse (teaching multiple students are better than teaching one student).

**Strengths:**

- The paper is exceedingly well-written and structured, making it easy to follow for a reader. The diagrams are informative and helpful.
- The idea is simple and brilliant -- it could be could be very effective for LLM finetuning and is executed well.
- Particularly M3 is reminiscent of ideas from OpenAI (AlphaFold). This is a very interesting exploration.
- The experiments are plentiful and convincing.

**Weaknesses:**

Figure 4, Table 2, Table 3, and Table 4 must include at least standard error or ideally 95% CI to prove statistical significance of the results.

With the number of acronyms in table 3 metrics, it's hard to understand what exactly numbers mean. Please find some way to include more understandable question categorizations.

The paper is missing literature from the learning sciences backing up the authors' strategy of implementing feedback and their premise of learning-by-teaching. Including a discussion on this would be important for motivating their work. Please examine literature on feedback (i.e. Power of Feedback, Hattie et al. or from the recent EDM, AIED, LAK communities) to strengthen your discussion and motivation. Additionally, it would be useful to examine ways to evaluate teaching quality also from the same communities.

**Questions:**

What are the interpretability implications of teaching diverse students at different knowledge levels? A discussion on this might be useful in the paper.

Why choose the game theory + math reasoning datasets? The motivation for the dataset choice seems to be a bit missing in the paper.

Can this be extended to open-ended math problems? How and what would need to change in the architecture?

**Limitations:**

The limitations are discussed mostly in terms of extensions. A discussion of bias perpetuation along the student / teacher pipeline i.e. Wambsganss et al. "Bias at a Second Glance" (COLING 2022) and global interpretability implications would be helpful here.

---

> ### Author Rebuttal · Authors · 2024-08-07
>
> **Q1: Figure 4, Table 2, Table 3, and Table 4 must include at least standard error or ideally 95% CI to prove statistical significance.**
>
> Thanks for this valuable suggestion. Due to the high cost of running experiments with LLMs, we reported the standard errors using the "bootstrapping" method [1, 2] in Figure 4 (Left) and Table 6 (Appendix A.1.3). We will clarify this in the revision. Specifically, in Figure 4, standard errors were calculated for K < 128 (number of TR-TA pairs) by selecting K pairs from the total 128 TR-TA pairs. Similarly, standard errors are presented for K = 12 in Table 6, showing that M1 with K = 12 can outperform SC with K=128.
>
> Following the suggestion, we extend the M1 Math experiments in Table 2 and Figure 4 to include up to 256 TR-TA pairs, enabling us to compute standard errors for the original 128-pair setting. We have also included additional experiments for M2 with 3 repeated runs. Please refer to the updated results in the provided PDF.
>
> Nevertheless, we cannot obtain other standard errors through repeated runs during the rebuttal period due to the resource and time constraints. For instance, the experiments in Table 3 are primarily constrained by the rate limit of the LeetCode server.
>
> **Q2: With the number of acronyms in table 3 metrics, it's hard to understand what exactly numbers mean. Please find some way to include more understandable question categorizations.**
>
> Thanks for pointing this out. There is a typo in our original caption stating that each acronym represents a question category. In fact, each acronym in the M1-Code experiments, such as SG-1 or SG-2, represents an individual question. All results in Table 3 are from the Game Theory category. More broadly, we experimented with three general question categories: Game Theory, Bitmasking, and General-1D, and summarized their results in Tables 3/9, Tables 10/11, and Tables 12/13, respectively.
>
> We will correct the typo in the caption and clarify the settings in the revision. Please do not hesitate to let us know if other clarification is needed for Table 3.
>
> **Q3: The paper is missing literature from the learning sciences backing up the authors' strategy of implementing feedback and their premise of LbT. Including a discussion on this would be important for motivating their work. Please examine literature on feedback to strengthen your discussion and motivation. Additionally, it would be useful to examine ways to evaluate teaching quality also from the same communities.**
>
> Thanks for this valuable suggestion. We agree that discussing more literature from the learning sciences will not only strengthen the motivation of our current implementation but also provide valuable insights for further improving the methods. We will add the discussions to our revision.
>
> To back up our concrete implementation of the LbT idea, we review several studies within the field [3, 4, 5, 6, 7, 8, 9, 10] that discuss how students' feedback can enhance a teacher's capabilities. The benefits of such feedback can be attributed to:
> - Reflection [7, 11, 12]: Teachers monitor and reflect on how well their ideas are understood by students and this reflection aids in evaluating their own understanding of domain concepts.
> - Knowledge-building [4, 5, 6]: Through interactions with students, such as questioning, teachers reflect upon their own expertise and comprehension, and become aware of their own misconceptions, and then attempt to repair them.
>
> Reflection directly supports the design of M1 and M3, where teachers improves their answer quality by observing how well a student answers similar problems (in M1) or reflecting on failure cases from multiple students (in M3). Additionally, knowledge-building backs up the implementation of M2, where students' feedback helps a teacher identify misconceptions, which are then addressed through DPO.
>
> For the roadmap of further improving the LbT implementation, Section 6.3 currently discusses our perspective on the general LbT pipeline, including potentially useful strategies for teaching material design and educational pipeline design. We will extend this section and discuss relevant literature from the learning sciences, based on the framework in Figure 7, focusing on the following points:
> - "cooperative learning" [13] implies dividing a difficult topic (as mentioned in "Task-oriented collaborative learning", Section 6.3) into several specific topics so that multiple agents can learn jointly.
> - "teachable agents" [7, 9, 10] indicate that appropriately configuring a student's knowledge level enables it to provide more useful feedback to the teacher.
> - Regarding the suggestion to "examine ways to evaluate teaching quality", we note that feedback can take many forms [14, 15]. Besides taking tests or exams [4, 5, 6], students can provide valuable feedback through their perception, such as peer recommendations [11] and satisfaction questionnaires [16, 17]. This implies that students could offer free-form feedback, which may be especially beneficial for open-ended problems.

---

> ### Author Response · Authors · 2024-08-07
> **Additional rebuttal 1**
>
> **Q4: What are the interpretability implications of teaching diverse students at different knowledge levels?**
>
> Thanks for bringing up this interesting topic. If our understanding of "interpretability" below does not align with your thoughts, we are happy to provide further clarifications during the discussion phase.
>
> We observed that teaching diverse students at different knowledge levels contributes positively (Table 2, 3, 5).
>
> For M3, the teacher makes verbalized reflections on why the current in-context learning examples are causing students' mistakes. During the rebuttal period, we conducted further analyses on the task of identifying false generalization fallacies [18] to verify that these natural language reflections could help interpret (1) how student diversity helps and (2) in-context learning behaviors.
>
> For (1), the teacher identifies **diverse and complementary causes** (numbered a,b,c) of errors from mistakes of different students (numbered 1,2,3), as listed in Table 3 of the attached PDF, which **help interpret why having diverse students is better**.
>
> We verified that the causes of students' mistakes (1a, 1b, ...) are indeed also causes of teachers' mistakes. Specifically, for each mistake the teacher makes on the test set, we prompt an LLaMa-3-70B to judge which cause categories (1a, 1b,...) does this mistake falls into. Note that one mistake can be caused by multiple causes simultaneously. Then, we report the percentage of teacher mistakes of that cause in Table 3's "% teacher mistakes of the same cause" column. By choosing a student model different from the teacher model, we identify more types of valid causes of teacher mistakes.
>
> Finally, these causes in Table 3 indeed help the teacher improve the ICL examples. After the teacher revises the ICL examples by learning from student 1, the teacher's mistakes caused by 1a, 1b & 1c are reduced by 6.0%, 11.6%, and 13.3%. Based on this, we conjecture that LbT methods like M3 could lead to a more interpretable way to understand the flaws of in-context learning models.
>
> **Q5: Why choose the game theory + math reasoning datasets?**
>
> We regard the ability of **accurate knowledge and reasoning** as the most crucial for advancing the capabilities and broad applications of LLMs. Drawing from human learning experiences, the LbT methodology has proven effective in providing intrinsic supervision for accurate knowledge building and reasoning. Our work aims to evaluate whether LbT can similarly benefit contemporary LLMs.
>
> Therefore, we choose math reasoning and competition-level code synthesis (including game theory, bitmasking, and general-1D) because they require **accurate knowledge and reasoning** and cannot be effectively solved with vague logic or reciting. Besides, these tasks are both popular and challenging, attracting considerable attention from the community. Successfully applying our approach to these tasks would demonstrate its effectiveness and highlight its potential for a wide-ranging impact.
>
> That being said, LbT could potentially benefit other tasks and datasets as well. We leave it to future work.
>
> **Q6: Can this be extended to open-ended math problems? How and what would need to change in the architecture?**
>
> Thanks for this valuable question. LbT can indeed be extended to open-ended problems, such as dialogue, writing, and open-ended math problems. A natural extension could involve using a teacher LLM to evaluate a student's answer, which would then serve as the LbT score. This approach mirrors how a teacher assesses a student in human learning and is promising due to the strong evaluation capabilities of LLMs [19, 20]. A similar idea has been employed to generalize Self-Consistency (which only applies to verifiable problems) [21] to open-form problems [22].
>
> Besides, as demonstrated in literature from the learning sciences, teachers may directly benefit from providing assessments [8] or reviews [23], and students can provide valuable free-form feedback beyond taking tests or exams, such as peer recommendations [11] and satisfaction questionnaires [16, 17]. We believe that an open-ended evaluation process could potentially offer additional feedback to the teacher LLM, which can be utilized to further enhance its performance. We plan to explore these extensions in future studies.

---

> ### Author Response · Authors · 2024-08-07
> **Additional rebuttal 2**
>
> **Q7: A discussion of bias perpetuation along the student / teacher pipeline i.e. Wambsganss et al. "Bias at a Second Glance" (COLING 2022) and global interpretability implications would be helpful here.**
>
> Thanks for raising this worth-discussing topic. In open-domain problems where no ground truth judgment exists (and LLM-based judgment might be needed), it is possible that teaching materials that are "well accepted and learned" by students may not necessarily be more accurate or closer to the truth, but may instead align with the existing biases of teachers or students. This poses a risk of the teacher perpetuating their own biases or indirectly learning the students' biases.
>
> Moreover, while our work primarily focuses on leveraging LbT for mathematical and code reasoning abilities, the importance of addressing these biases is even greater in domains where societal bias and fairness are significant concerns.
>
> We will add a discussion in the revision as this topic should be carefully considered in future work.

---

> > ### Comment · Reviewer_UmKq · 2024-08-11
> >
> > Hello authors,
> >
> > Thank you for your rebuttal! I would like to point out one concern: the rebuttal is strictly supposed to be 6000 characters. Adding not one but two additional comments to address concerns is disrespectful to both my time as a reviewer but more importantly disrespectful to other authors who worked very diligently in reducing their rebuttal to the word limit. I would like the PCs and Senior ACs to suggest a policy of what to do in this situation.
> >
> > Thank you for the additional experiments. I would request that in Table 3, you could produce std devs. by choosing a smaller sample size or simply bootstrap sampling.
> >
> > For Q3-Q7, could you suggest exactly what parts of the rebuttal discussion will be included in the paper and in what sections? I feel all these points are important to touch upon.

---

> > > ### Author Response · Authors · 2024-08-12
> > > **Thanks for your reply and further response from the authors**
> > >
> > > Dear reviewer,
> > >
> > > Thanks for the reply and follow-up questions. We answer them as follows:
> > >
> > > > Adding not one but two additional comments to address concerns is disrespectful.
> > >
> > > Thanks for bringing up this concern to us directly. We are really sorry for any feelings of disrespect you have experienced. Our intention was never to violate any rules or show disrespect. We are used to using multiple comments as in previous conferences, which makes us overlook the need to shorten the question quotation and answers. Setting intentions aside, it is our bad that raises your concern, and we fully respect and accept any decisions of you and PCs/SACs.
> > >
> > > > I would request that in Table 3, you could produce std devs. by choosing a smaller sample size or simply bootstrap sampling.
> > >
> > > Following the suggestion, we use bootstrap sampling (choose 4 TR-TAs from 8 TR-TAs, 20 sets are sampled) to produce a table with standard deviation as follows:
> > >
> > > | Models | Metrics | SG-1 | SG-2 | SG-3 | SG-4 | PW |
> > > | - | - | - | - | - | - | - |
> > > |   | Avg. | 0.198&plusmn;0.125 | 0.004&plusmn;0.002 | 0.209&plusmn;0.068 | 0.564&plusmn;0.084 | 0.613&plusmn;0.128
> > > | T=LLaMA3-8B | M1(Max) | 0.539&plusmn;0.162 | 0.004&plusmn;0.004 | 0.220&plusmn;0.116 | 0.690&plusmn;0.233 | 0.576&plusmn;0.346
> > > | S=LLaMA3-8B | Avg.(V=1) | 1.000&plusmn;0.000 | - | - | 0.647&plusmn;0.294 | 0.847&plusmn;0.113
> > > |   | M1(Max)(V=1) | 1.000&plusmn;0.000 | - | - | 0.724&plusmn;0.332 | 0.939&plusmn;0.124
> > > |:-:|:-:|:-:|:-:|:-:|:-:|:-:|
> > > |   | Avg. | 0.335&plusmn;0.142 | 0.005&plusmn;0.002 | 0.292&plusmn;0.108 | 0.591&plusmn;0.106 | 0.695&plusmn;0.076
> > > | T=LLaMA3-8B | M1(Max)  | 0.382&plusmn;0.242 | 0.009&plusmn;0.004 | 0.503&plusmn;0.130 | 0.728&plusmn;0.154 | 0.723&plusmn;0.121
> > > | S=LLaMA3-8B | Avg.(V=1) | 0.827&plusmn;0.147 | - | - | 0.653&plusmn;0.318 | 0.890&plusmn;0.104
> > > | (Self-debug)| M1(Max)(V=1) | 0.928&plusmn;0.196 | - | - | 0.815&plusmn;0.346 | 0.941&plusmn;0.072
> > > |:-:|:-:|:-:|:-:|:-:|:-:|:-:|
> > > |   | Avg. | 0.582&plusmn;0.128 | 0.007&plusmn;0.002 | 0.432&plusmn;0.177 | 1.000&plusmn;0.000 | 0.643&plusmn;0.132
> > > | T=GPT-3.5 | M1(Max)  | 0.827&plusmn;0.178 | 0.010&plusmn;0.002 | 0.631&plusmn;0.193 | 1.000&plusmn;0.000 | 0.774&plusmn;0.129
> > > | S=GPT-3.5 | Avg.(V=1) | 0.993&plusmn;0.006 | - | 0.746&plusmn;0.299 | 1.000&plusmn;0.000 | 0.914&plusmn;0.082
> > > |  | M1(Max)(V=1) | 1.000&plusmn;0.000 | - | 0.593&plusmn;0.432 | 1.000&plusmn;0.000 | 0.962&plusmn;0.049
> > > |:-:|:-:|:-:|:-:|:-:|:-:|:-:|
> > > |   | Avg. | 0.723&plusmn;0.162 | 0.096&plusmn;0.119 | 0.586&plusmn;0.136 | 1.000&plusmn;0.000 | 0.841&plusmn;0.070
> > > | T=GPT-3.5 | M1(Max)  | 1.000&plusmn;0.000 | 0.255&plusmn;0.368 | 0.655&plusmn;0.323 | 1.000&plusmn;0.000 | 0.911&plusmn;0.104
> > > | S=GPT-3.5 | Avg.(V=1) | 0.996&plusmn;0.004 | 1.000&plusmn;0.000 | 0.666&plusmn;0.338 | 1.000&plusmn;0.000 | 0.897&plusmn;0.088
> > > | (Self-debug)| M1(Max)(V=1) | 1.000&plusmn;0.000 | 1.000&plusmn;0.000 | 0.666&plusmn;0.338 | 1.000&plusmn;0.000 | 0.931&plusmn;0.075
> > > |:-:|:-:|:-:|:-:|:-:|:-:|:-:|
> > > |   | Avg. | 0.838&plusmn;0.119 | 0.008&plusmn;0.001 | 0.677&plusmn;0.135 | 1.000&plusmn;0.000 | 0.597&plusmn;0.102
> > > | T=LLaMA3-70B | M1(Max)  | 0.900&plusmn;0.200 | 0.007&plusmn;0.002 | 0.787&plusmn;0.398 | 1.000&plusmn;0.000 | 0.671&plusmn;0.112
> > > | S=LLaMA3-8B | Avg.(V=1) | 1.000&plusmn;0.000 | - | 1.000&plusmn;0.000 | 1.000&plusmn;0.000 | 0.918&plusmn;0.127
> > > |  | M1(Max)(V=1) | 1.000&plusmn;0.000 | - | 1.000&plusmn;0.000 | 1.000&plusmn;0.000 | 0.965&plusmn;0.112
> > >
> > > > For Q3-Q7, could you suggest exactly what parts of the rebuttal discussion will be included in the paper and in what sections? I feel all these points are important to touch upon.
> > >
> > > Q3: We will merge the discussion into the current Section 6.3 ("Borrowing Education Strategies to Improve LLMs"), including both the literature supporting the design and the literature supporting our prospects in Figure 7.
> > >
> > > Q4: We will add a subsection under Appendix C ("M3") to provide this detailed interpretation of M3's working process.
> > >
> > > Q5: We will add the discussion on this important experimental design choice to Appendix A.1.2 ("Additional Experimental Setups"), and refer to it at the end of Section 3.1.
> > >
> > > Q6: We will add these two sentences "LbT can potentially be extended to open-ended problems, such as dialogue, writing, and open-ended math problems. A natural extension could involve using a teacher LLM to evaluate a student's answer, which would then serve as the LbT score" to Section 6.1 ("Limitations and Near-Term Extensions").
> > >
> > > Q7: We will add this important discussion to a standalone section in Section 6, positioned between the current Section 6.1 and 6.2. The title will be "Potential Risk of Bias Perpetuation".
> > >
> > > Best,
> > >
> > > Authors

---

> > > > ### Comment · Reviewer_UmKq · 2024-08-12
> > > >
> > > > Thanks! I've raised my score to accept.

---

> > > > > ### Author Response · Authors · 2024-08-13
> > > > >
> > > > > Thank you for all your valuable suggestions, understanding, and support! We'll make sure to incorporate the suggestions.

---

### Author Rebuttal · Authors · 2024-08-07

We sincerely thank all the reviewers for their valuable time and effort in reviewing our paper. We are encouraged that the reviewers recognize our paper as novel and interesting (UmKq, mNjM); see its potential impact on the LLM community (UmKq, mNjM); note the abundance of experiments included (UmKq, mNjM, E9eU); and find it well-written and easy to follow (UmKq, pWvs, mNjM). We also appreciate the thoughtful concerns and suggestions, which are both inspiring and valuable for further discussion.

In response to the comments, we have added several new experiments which can be found in the attached PDF:
- We have extended the number of TR-TA pairs in M1 Math to 256 (Table 1 in the PDF).
- We have included additional experiments for M1 Math with LLaMA3-70B teaching Mistral-7B and LLaMA3-70B teaching LLaMA3-8B + Mistral-7B (Table 1 and Table 2 in the PDF).
- We have updated the plot showing the relative improvements of M1 over SC with up to 256 TR-TA pairs (Figure 1 in the PDF).
- We have calculated standard errors for M1 Math with K < 256 (number of TR-TA pairs), by selecting K pairs from the total 256 TR-TA pairs (Table 2 and Figure 1 in the PDF).
- We have calculated standard errors for M2 with 3 repeated runs (Table 3 in the PDF).
- We have added an analysis of the causes of errors identified by the teacher (LLaMa3-70B) in M3 (Table 4 in the PDF).

**References**

[1] Math-Shepherd: Verify and Reinforce LLMs Step-by-step without Human Annotations. ACL, 2024.

[2] Improve mathematical reasoning in language models by automated process supervision. arXiv, 2024.

[3] Children teach children: Learning by teaching. Harper & Row, 1971.

[4] The influence of the tutee in learning by peer tutoring. Proceedings of the Annual Meeting of the Cognitive Science Society, 2004.

[5] Understanding tutor learning: Knowledge-building and Knowledge-telling in Peer Tutors' Explanations and Questions. Review of
Educational Research, 2007.

[6] Tutor learning: The Role of Explaining and Responding to Questions. Instructional Science, 2008.

[7] Learning by teaching: A New Agent Paradigm for Educational Software. Applied Artificial Intelligence, 2005.

[8] Learning-by-teaching. Evidence and implications as a pedagogical mechanism. Innovations in Education and Teaching International,
2017.

[9] Can you clarify what you said?: Studying the Impact of Tutee Agents' Follow-up Questions on Tutors' Learning. AIED, 2021.

[10] Teach AI how to code: Using Large Language Models as Teachable Agents for Programming Education. CHI, 2024.

[11] Building a metacognitive model of reflection. Higher Eduction, 1999.

[12] Learning from human tutoring. Cognitive Science, 2001.

[13] Evaluation of Jigsaw, a cooperative learning technique. Contemporary Educational Psychology, 1985.

[14] The power of feedback. Review of Educational Research, 2007.

[15] The power of feedback revisited: A Meta-analysis of Educational Feedback Research. Frontiers in Psychology, 2020.

[16] Quantifying quality: The Importance of Student Feedback. Quality In Higher Education, 2001.

[17] Instruments for obtaining student feedback: A Review of the Literature. Assessment & Evaluation in Higher Education, 2005.

[18] Logical fallacy detection. EMNLP Findings, 2022.

[19] Judging LLM-as-a-judge with MT-Bench and chatbot arena. NeurIPS, 2023.

[20] From crowdsourced data to high-quality benchmarks: Arena-Hard and BenchBuilder Pipeline. arXiv, 2024.

[21] Self-consistency improves chain of thought reasoning in language models. ICLR, 2023.

[22] Universal self-consistency for large language model generation. arXiv, 2023.

[23] Learning by reviewing. Journal of Educational Psychology, 2011.

[24] Chain-of-thought prompting elicits reasoning in large language models. NeurIPS, 2022.

[25] Self-refine: Iterative Refinement with Self-Feedback. NeurIPS, 2023.

[26] Reflexion: Language Agents with Verbal Reinforcement Learning. NeurIPS, 2023.

---

### Decision · Program_Chairs · 2024-09-25

**Decision:**

Accept (poster)

**Comment:**

The reviewers agreed that the paper investigates a novel setting of how learning-by-teaching ideas can help improve LMMs and that the results would be of broad interest to the community. However, the reviewers also raised several concerns and questions in their initial reviews. We want to thank the authors for their responses and active engagement during the discussion phase. The reviewers appreciated the responses, which helped in answering their key questions. The reviewers have an overall positive assessment of the paper, and there is a consensus for acceptance. The reviewers have provided detailed feedback, and we strongly encourage the authors to incorporate this feedback when preparing the final version of the paper.